# Enhancing Prediction Performance through Influence Measure

**Shuguang Yu**[a],[*] **Wenqian Xu**[a], **Xinyi Zhou**[a], **Xuechun Wang**[a], **Hongtu Zhu**[b], and **Fan Zhou**[a],[†]
[a]Shanghai University of Finance and Economics, Shanghai, China
[b]University of North Carolina at Chapel Hill, North Carolina, USA

## Abstract

In the field of machine learning, the pursuit of accurate models is ongoing. A key aspect of improving prediction performance lies in identifying which data points in the training set should be excluded and which high-quality, potentially unlabeled data points outside the training set should be incorporated to improve the model's performance on unseen data. To accomplish this, an effective metric is needed to evaluate the contribution of each data point toward enhancing overall model performance. This paper proposes the use of an influence measure as a metric to assess the impact of training data on test set performance. Additionally, we introduce a data selection method to optimize the training set as well as a dynamic active learning algorithm driven by the influence measure. The effectiveness of these methods is demonstrated through extensive simulations and real-world datasets.

## 1 Introduction

To build effective machine learning models, the significance of individual training data points cannot be overstated. Each data point in the training set contributes uniquely to the model's learning process, shaping its performance, generalization, and resilience to various challenges (Blum & Langley, 1997). Simply evaluating the model's performance on the provided data is insufficient; understanding the influence of individual training examples and making informed decisions about their inclusion or exclusion is critical for developing effective and reliable models.

Recent advancements in machine learning have highlighted the importance of strategic data selection and management during training. Techniques such as active learning (Settles, 2009) promote the iterative selection of the most uncertain unlabeled data points for labeling and inclusion in the training set. However, estimating uncertainty in deep neural networks (DNNs) is challenging due to their tendency to exhibit overconfidence (Ren et al., 2021). To address this, methods like deep Bayesian approaches (Gal et al., 2017), query-by-committee (Gorriz et al., 2017), Variational Auto-Encoders (Sinha et al., 2019), adversarial learning (Ducoffe & Precioso, 2018; Mayer & Timofte, 2020), graph convolutional networks (Caramalau et al., 2021), and noise stability (Li et al., 2024) have been proposed to improve uncertainty estimates. However, these approaches assume a well-trained model and fail to consider how model parameters might evolve when the training data is modified.

To bridge this gap, recent studies have focused on quantifying the impact of individual training examples on model behavior, with the main challenge being the identification of an appropriate evaluation metric. The Shapley value has emerged as a promising solution, inspiring a number of Shapley-value-based approaches (Ghorbani & Zou, 2019; Jia et al., 2019a;b; Ghorbani et al., 2020; Kwon & Zou, 2022; Wang & Jia, 2023). However, these methods often require multiple model retrainings and evaluations, making them computationally expensive.

Techniques such as influence functions (Koh & Liang, 2017; Pruthi et al., 2020; Yeh et al., 2018; Chen et al., 2021) offer insights into the effect of individual data points on model predictions, helping to identify and mitigate harmful or overly influential examples. For instance, Chhabra et al. (2024) apply influence functions to measure the impact of training data on a specific model, improving

---

[*]The first four authors have contributed equally to this paper.
[†]Corresponding author. Email: zhoufan@mail.shufe.edu.cn

performance by pruning detrimental data points. Their method is highly efficient as it avoids the need for model retraining. However, these algorithms may still fail in certain cases, as demonstrated in the following simple example.

**Binary Classification via Logistic Regression.** We consider a logistic regression model $p(x) = (1 + e^{-x\beta})^{-1}$, where $\beta \in \mathbb{R}$ represents the coefficients. For a training set $\mathcal{Z} = \{(x_i, y_i)\}_{i=1}^{n}$ and a validation set $\mathcal{V}$, the influence function in Chhabra et al. (2024) for example is defined as

$$\mathcal{I}(-x_i) = \sum_{(x,y) \in \mathcal{V}} \partial_\beta L(y, x; \hat{\beta}) [\sum_{i=1}^{n} \partial_\beta^2 L(y_i, x_i; \hat{\beta})]^{-1} \partial_\beta L(y_i, x_i; \hat{\beta}),$$

where $\partial_\beta = \partial/\partial\beta$, $x \in \mathbb{R}$, $y \in \{0, 1\}$ is a binary classification label, $L$ represents the cross-entropy loss, and $\hat{\beta} = \arg\min_{\beta \in \mathbb{R}} n^{-1} \sum_{i=1}^{n} L(y_i, x_i; \beta)$. According to Chhabra et al. (2024), samples $x_i$ with negative influence function values negatively impact the model's performance on the validation set and should be removed. For all training samples $x_i$, the term $\sum_{(x,y) \in \mathcal{V}} \partial_\beta L(y, x; \hat{\beta}) [\sum_{i=1}^{n} \partial_\beta^2 L(y_i, x_i; \hat{\beta})]^{-1}$ remains constant, with only $\partial_\beta L(y_i, x_i; \hat{\beta})$ varying. In the simple logistic regression scenario, we have $\partial_\beta L(y_i, x_i; \hat{\beta}) = [(1 + e^{-x_i\hat{\beta}})^{-1} - y_i]x_i$. Assuming $\sum_{(x,y) \in \mathcal{V}} \partial_\beta L(y, x; \hat{\beta}) [\sum_{i=1}^{n} \partial_\beta^2 L(y_i, x_i; \hat{\beta})]^{-1} > 0$ and $x_i > 0$, the training sample influence is negative if and only if $y_i = 1$. However, it is clearly incorrect to solely remove data points from one class, as outliers in the other class may also negatively influence the model's performance. For higher-dimensional $x_i$, as shown in Section 6 using both simulated and real-world data, our method consistently outperforms the approach proposed by Chhabra et al. (2024) and others.

Given the limitations of existing methods, we draw inspiration from local influence measures developed in the statistical community (Zhu et al., 2007; 2011; Shu & Zhu, 2019; Sui et al., 2023) to propose a novel metric in this article. Our approach differs from previous ones that typically evaluate the impact of perturbations on data samples or model parameters with a fixed model assumption. Instead, our measure directly assesses the effect of minor perturbations to training samples on the model's performance on a validation set, allowing the model to adapt to these changes. To implement this, we have developed a new perturbation manifold and expanded the local influence framework. Using this innovative approach, we introduce two key metrics: one for data trimming, aimed at identifying and removing training set anomalies that compromise model performance on the test set, and another for active learning, which focuses on selecting the most impactful unlabeled data to enhance the prediction performance. Moreover, acknowledging the challenges of slow computation and high memory usage inherent in calculating exact local influence measures, we propose two approximation methods. Our experiments on real-world datasets demonstrate that these approximations achieve performance comparable to exact calculations while significantly reducing computational overhead.

We summarize our contributions as follows,

(i) Unlike existing local influence measures, we propose a new metric that evaluates the impact of perturbations to training samples on the model's performance on validation data.

(ii) The proposed metric is applicable to both data trimming and active learning. For data trimming, it evaluates the effects of minor perturbations to each sample, offering deeper insights into how individual samples impact model performance. In the context of active learning, the method captures the relationship between training samples, unlabeled data, and parameter updates.

(iii) We propose two approximation methods to alleviate the high computational cost of calculating local influence measures. These algorithms significantly reduce computational overhead while maintaining better performance than other methods, as demonstrated in our experiments.

## 2 A New Influence Measure

In this section, we begin by presenting essential background information, including the Perturbation Manifold, before formally defining the proposed metric.

**Perturbation Manifold.** Our definition of the perturbation manifold closely follows that of Shu & Zhu (2019). Given an input sample $z = (x, y)$ in the training set $\mathcal{Z} = \{z_i\}_{i=1}^{n}$ and a machine

learning model with an estimated parameter vector $\hat{\boldsymbol{\theta}}$, which is trained on $\mathcal{Z}$, the prediction probability for class $c \in \{1, \ldots, K\}$ is denoted as $P(c|\boldsymbol{x}, \hat{\boldsymbol{\theta}})$. Let $\boldsymbol{\omega} = (\omega_1, \ldots, \omega_p)^\top$ be a perturbation vector that varies within an open subset $\Omega \subset \mathbb{R}^p$. The perturbation $\boldsymbol{\omega}$ is applied to $\boldsymbol{x}$, thereby affecting the learning of the parameter vector $\hat{\boldsymbol{\theta}}$. We denote the parameter vector obtained by the model after perturbing the training sample $\boldsymbol{x}$ with $\boldsymbol{\omega}$ as $\hat{\boldsymbol{\theta}}(\boldsymbol{x}+\boldsymbol{\omega})$ with $\hat{\boldsymbol{\theta}}(\boldsymbol{x}) = \hat{\boldsymbol{\theta}}$. We define $P(c|\boldsymbol{x}+\boldsymbol{\omega}, \hat{\boldsymbol{\theta}}(\boldsymbol{x}+\boldsymbol{\omega}))$ as the prediction probability under the perturbation $\boldsymbol{\omega}$ such that $\sum_{c=1}^K P(c|\boldsymbol{x} + \boldsymbol{\omega}, \hat{\boldsymbol{\theta}}(\boldsymbol{x} + \boldsymbol{\omega})) = 1$. It is assumed that there exists a $\boldsymbol{\omega}_0 \in \Omega$ such that $P(c|\boldsymbol{x} + \boldsymbol{\omega}_0, \hat{\boldsymbol{\theta}}(\boldsymbol{x} + \boldsymbol{\omega}_0)) = P(c|\boldsymbol{x}, \hat{\boldsymbol{\theta}})$. Additionally, we assume that $\{P(c|\boldsymbol{x} + \boldsymbol{\omega}, \hat{\boldsymbol{\theta}}(\boldsymbol{x} + \boldsymbol{\omega}))\}_{c=1}^K$ is positive and sufficiently smooth for all $\boldsymbol{\omega} \in \Omega$.

Following the development in (Zhu et al., 2007; 2011), we define $\mathcal{M} = \{P(c|\boldsymbol{x} + \boldsymbol{\omega}, \hat{\boldsymbol{\theta}}(\boldsymbol{x} + \boldsymbol{\omega})) : \boldsymbol{\omega} \in \Omega\}$ as a perturbation manifold. The tangent space of $\mathcal{M}$ at $\boldsymbol{\omega}$ is denoted by $T_{\boldsymbol{\omega}}$, which is spanned by $\{\partial l(\boldsymbol{\omega}|c, \boldsymbol{x}, \hat{\boldsymbol{\theta}}(\boldsymbol{x}))/\partial \omega_i\}_{i=1}^p$, where $l(\boldsymbol{\omega}|c, \boldsymbol{x}, \hat{\boldsymbol{\theta}}(\boldsymbol{x})) = \log P(c|\boldsymbol{x} + \boldsymbol{\omega}, \hat{\boldsymbol{\theta}}(\boldsymbol{x} + \boldsymbol{\omega}))$. Let $\boldsymbol{G}_{\boldsymbol{z}}(\boldsymbol{\omega}) = \sum_{c=1}^K \partial_{\boldsymbol{\omega}}^\top l(\boldsymbol{\omega}|c, \boldsymbol{x}, \hat{\boldsymbol{\theta}}(\boldsymbol{x})) \partial_{\boldsymbol{\omega}} l(\boldsymbol{\omega}|c, \boldsymbol{x}, \hat{\boldsymbol{\theta}}(\boldsymbol{x})) P(c|\boldsymbol{x} + \boldsymbol{\omega}, \hat{\boldsymbol{\theta}}(\boldsymbol{x} + \boldsymbol{\omega}))$ with $\partial_{\boldsymbol{\omega}} = (\partial/\partial \omega_1, \ldots, \partial/\partial \omega_p)$. If $\boldsymbol{G}_{\boldsymbol{z}}(\boldsymbol{\omega})$ is positive definite, then $\mathcal{M}$ is a Riemannian manifold (Shu & Zhu, 2019) with $\boldsymbol{G}_{\boldsymbol{z}}(\boldsymbol{\omega})$ serving as the Riemannian metric tensor (Amari, 2012; Amari & Nagaoka, 2000).

Although $\boldsymbol{G}_{\boldsymbol{z}}(\boldsymbol{\omega})$ is often not positive definite in classification problems, we can still reduce the dimensionality of the perturbations and reconstruct a Riemannian manifold (Shu & Zhu, 2019).

**The Influence Measure.** Let $L(y', \boldsymbol{x}'; \boldsymbol{\theta})$ denote the loss function of the model with parameter $\boldsymbol{\theta}$ on $\boldsymbol{z}' = (\boldsymbol{x}', y') \notin \mathcal{Z}$, we can get the expression for the (first-order) influence measure:

$$\text{FI}(\boldsymbol{z}', \boldsymbol{z}) = \partial_{\boldsymbol{\omega}} L(y', \boldsymbol{x}'; \hat{\boldsymbol{\theta}}(\boldsymbol{x} + \boldsymbol{\omega}_0)) \boldsymbol{G}_{\boldsymbol{z}}^\dagger(\boldsymbol{\omega}_0) \partial_{\boldsymbol{\omega}}^\top L(y', \boldsymbol{x}'; \hat{\boldsymbol{\theta}}(\boldsymbol{x} + \boldsymbol{\omega}_0)), \quad (2.1)$$

where $\boldsymbol{G}_{\boldsymbol{z}}^\dagger(\boldsymbol{\omega}_0)$ is the pseudoinverse of $\boldsymbol{G}_{\boldsymbol{z}}(\boldsymbol{\omega}_0)$.

We consider a linear perturbation approach where applying perturbation $\boldsymbol{\omega}$ to the training sample $\boldsymbol{z}$ transforms $(\boldsymbol{x}, y)$ into $(\boldsymbol{x} + \boldsymbol{\omega}, y)$. Consequently, the parameter vector updates to $\hat{\boldsymbol{\theta}}(\boldsymbol{x} + \boldsymbol{\omega}) := n^{-1} \arg\min_{\boldsymbol{\theta} \in \Theta} \{\sum_{i=1}^n L(y_i, \boldsymbol{x}_i; \boldsymbol{\theta}) + L(y, \boldsymbol{x} + \boldsymbol{\omega}; \boldsymbol{\theta}) - L(y, \boldsymbol{x}; \boldsymbol{\theta})\}$, with $\boldsymbol{\omega}_0 = \mathbf{0}$. Using the chain rule, we can derive the expression for $\partial_{\boldsymbol{\omega}} L(y', \boldsymbol{x}'; \hat{\boldsymbol{\theta}}(\boldsymbol{x} + \boldsymbol{\omega}_0))$:

$$\partial_{\boldsymbol{\omega}} L(y', \boldsymbol{x}'; \hat{\boldsymbol{\theta}}(\boldsymbol{x} + \boldsymbol{\omega}_0)) = \partial_{\boldsymbol{\theta}} L(y', \boldsymbol{x}'; \hat{\boldsymbol{\theta}}(\boldsymbol{x})) \partial_{\boldsymbol{\omega}} \hat{\boldsymbol{\theta}}(\boldsymbol{x} + \boldsymbol{\omega})\big|_{\boldsymbol{\omega} = \boldsymbol{\omega}_0}. \quad (2.2)$$

Here, $\partial_{\boldsymbol{\omega}} \hat{\boldsymbol{\theta}}(\boldsymbol{x} + \boldsymbol{\omega})\big|_{\boldsymbol{\omega} = \boldsymbol{\omega}_0} \approx n^{-1} \boldsymbol{H}_{\hat{\boldsymbol{\theta}}}^{-1} \partial_{\boldsymbol{x}} \partial_{\boldsymbol{\theta}} L(y, \boldsymbol{x}; \hat{\boldsymbol{\theta}}(\boldsymbol{x}))$, whose derivation is detailed in Appendix A), where $\boldsymbol{H}_{\hat{\boldsymbol{\theta}}} := n^{-1} \sum_{i=1}^n \partial_{\boldsymbol{\theta}}^2 L(y_i, \boldsymbol{x}_i; \hat{\boldsymbol{\theta}}(\boldsymbol{x}))$. The term $\partial_{\boldsymbol{x}} \partial_{\boldsymbol{\theta}} L(y, \boldsymbol{x}; \hat{\boldsymbol{\theta}}(\boldsymbol{x}))$ represents the gradient of the loss function $L$ first taken with respect to the model parameters $\boldsymbol{\theta}$ and then with respect to $\boldsymbol{x}$, evaluated at the perturbed training sample $(\boldsymbol{x}, y)$.

For the computation of $\boldsymbol{G}_{\boldsymbol{z}}(\boldsymbol{\omega}_0)$, $P(c|\boldsymbol{x} + \boldsymbol{\omega}_0, \hat{\boldsymbol{\theta}}(\boldsymbol{x} + \boldsymbol{\omega}_0))$ can be directly obtained using the learned model parameter vector $\hat{\boldsymbol{\theta}}$ and the unperturbed sample point $\boldsymbol{x}$. The calculation of $\partial_{\boldsymbol{\omega}} l(\boldsymbol{\omega}_0|c, \boldsymbol{x}, \hat{\boldsymbol{\theta}}(\boldsymbol{x}))$ requires the application of the chain rule:

$$\begin{aligned} \partial_{\boldsymbol{\omega}} l(\boldsymbol{\omega}_0|c, \boldsymbol{x}, \hat{\boldsymbol{\theta}}(\boldsymbol{x})) =& \partial_{\boldsymbol{\omega}} \log P(c|\boldsymbol{x} + \boldsymbol{\omega}_0, \hat{\boldsymbol{\theta}}(\boldsymbol{x} + \boldsymbol{\omega}_0)) \qquad\qquad (2.3) \\ =& \partial_{\boldsymbol{\theta}} \log P(c|\boldsymbol{x} + \boldsymbol{\omega}_0, \hat{\boldsymbol{\theta}}(\boldsymbol{x} + \boldsymbol{\omega}_0)) \cdot \partial_{\boldsymbol{\omega}} \hat{\boldsymbol{\theta}}(\boldsymbol{x} + \boldsymbol{\omega})|_{\boldsymbol{\omega} = \boldsymbol{\omega}_0} \\ &+ \partial_{\boldsymbol{x}} \log P(c|\boldsymbol{x} + \boldsymbol{\omega}_0, \hat{\boldsymbol{\theta}}(\boldsymbol{x} + \boldsymbol{\omega}_0)). \end{aligned}$$

All differentiation operations can be easily computed using backpropagation (Goodfellow et al., 2016) in deep learning libraries such as TensorFlow (Abadi et al., 2016) and PyTorch (Paszke et al., 2017). This entire process is efficient and does not require retraining the model parameters.

**Theorem 1.** If $\varphi$ represents a diffeomorphism of $\boldsymbol{\omega}$, then $\text{FI}(\boldsymbol{z}', \boldsymbol{z})$ is invariant under any reparameterization associated with $\varphi$.

Compared to widely used measures in Euclidean spaces, such as the Jacobian norm (Novak et al., 2018) and Cook's local influence measure (Cook, 2018), Theorem 1 demonstrates that $\text{FI}(\boldsymbol{z}', \boldsymbol{z})$ remains invariant under any diffeomorphic transformation (e.g., scaling) of the perturbation vector $\boldsymbol{\omega}$. The proof of Theorem 1 can be found in Shu & Zhu (2019).

The significance of Theorem 1 is especially pronounced when there are scale differences among the dimensions of $\boldsymbol{x}$. For instance, if certain dimensions have significantly larger values than others, the contribution of perturbations to those dimensions may appear exaggerated. However, our $\text{FI}(\boldsymbol{z}', \boldsymbol{z})$ mitigates this scaling issue by employing the metric tensor of the perturbation manifold instead of that of the standard Euclidean space.

## 3    FI FOR DATA TRIMMING.

The primary goal of data trimming is to eliminate training samples that may compromise the model's performance on datasets beyond the training set. Since our proposed influence measure (FI) quantifies the impact of each training sample on the model's performance on validation sets, it serves as a natural tool for data trimming.

From a model robustness perspective, if a small perturbation in a training sample leads to a significant effect on the model's performance on the validation set, that sample should be excluded, which aligns with the principle of our proposed metric. Given the inherent challenge of ensuring that all samples in the training set are entirely accurate, a sample with excessive influence could severely degrade the model's overall performance if it contains any contamination. Therefore, to enhance prediction performance on an unseen data set, such samples should be removed from the training set.

Following the setup of Chhabra et al. (2024), we introduce a training set $\mathcal{Z}$, a validation set $\mathcal{V}$, and a base model $\mathcal{F}$. In this context, we assess the impact of each training sample on the model's performance by computing the FI for each training tuple $z_i \in \mathcal{Z}$ with respect to $\mathcal{V}$. To achieve this, we extend Equation equation 2.1 to encompass the entire validation set. This involves replacing the loss function for an individual validation sample with the mean loss function across the entire validation set, as follows:

$$\mathrm{FI}^{util}(\boldsymbol{z}) = \partial_{\boldsymbol{\omega}} L(\mathcal{V}; \hat{\boldsymbol{\theta}}(\boldsymbol{x} + \boldsymbol{\omega}_0)) \boldsymbol{G}_{\boldsymbol{z}}^{\dagger}(\boldsymbol{\omega}_0) \partial_{\boldsymbol{\omega}}^{\top} L(\mathcal{V}; \hat{\boldsymbol{\theta}}(\boldsymbol{x} + \boldsymbol{\omega}_0)), \tag{3.1}$$

where $\partial_{\boldsymbol{\omega}} L(\mathcal{V}; \hat{\boldsymbol{\theta}}(\boldsymbol{x} + \boldsymbol{\omega}_0)) := \frac{1}{|\mathcal{V}|} \sum_{(\boldsymbol{x}', y') \in \mathcal{V}} \partial_{\boldsymbol{\omega}} L(y', \boldsymbol{x}'; \hat{\boldsymbol{\theta}}(\boldsymbol{x} + \boldsymbol{\omega}_0))$.

The algorithm for computing $\mathrm{FI}^{util}$ is outlined in Algorithm 1. After computing these values, we can sort all data points in the training set in descending order based on their $\mathrm{FI}^{util}$ values and remove the top $b$ points to enhance the model's performance on the test set. For the detailed data trimming algorithm, please refer to Algorithm 3.

---

**Algorithm 1** Calculation of $\mathrm{FI}^{util}$

---

**Input:** Training set $\mathcal{Z}$, Validation set $\mathcal{V}$, Base model $\mathcal{F}$
**Output:** Influence measure vector $\mathrm{FI}^{util} \in \mathbb{R}^{|\mathcal{Z}| \times 1}$

1: **procedure** $\mathrm{FI}^{util}$-CALCULATION($\mathcal{Z}, \mathcal{V}, \mathcal{F}$)
2:     Train $\mathcal{F}$ with $\mathcal{Z}$, and obtain the parameter vector $\hat{\boldsymbol{\theta}}$
3:     Generate an empty vector $\mathrm{FI}^{util}$ of size $|\mathcal{Z}| \times 1$
4:     Calculate $\frac{1}{|\mathcal{V}|} \sum_{(\boldsymbol{x}', y') \in \mathcal{V}} \partial_{\boldsymbol{\theta}} L(y', \boldsymbol{x}'; \hat{\boldsymbol{\theta}})$ and $\boldsymbol{H}_{\hat{\boldsymbol{\theta}}}$
5:     **for** every $\boldsymbol{z}_i$ in $\mathcal{Z}$ **do**
6:         Calculate $\boldsymbol{G}_{\boldsymbol{z}_i}(\boldsymbol{\omega}_0)$ and $\partial_{\boldsymbol{x}} \partial_{\boldsymbol{\theta}} L(y_i, \boldsymbol{x}_i; \hat{\boldsymbol{\theta}})$
7:         $\partial_{\boldsymbol{\omega}} L(\mathcal{V}; \hat{\boldsymbol{\theta}}) \leftarrow \left[ \frac{1}{|\mathcal{V}|} \sum_{(\boldsymbol{x}', y') \in \mathcal{V}} \partial_{\boldsymbol{\theta}} L(y', \boldsymbol{x}'; \hat{\boldsymbol{\theta}}) \right] \boldsymbol{H}_{\hat{\boldsymbol{\theta}}}^{-1} \partial_{\boldsymbol{x}} \partial_{\boldsymbol{\theta}} L(y_i, \boldsymbol{x}_i; \hat{\boldsymbol{\theta}})$
8:         $\mathrm{FI}^{util}[i] \leftarrow \partial_{\boldsymbol{\omega}} L(\mathcal{V}; \hat{\boldsymbol{\theta}}) \boldsymbol{G}_{\boldsymbol{z}_i}^{\dagger}(\boldsymbol{\omega}_0) \partial_{\boldsymbol{\omega}}^{\top} L(\mathcal{V}; \hat{\boldsymbol{\theta}})$
9:     **end for**
10:    **return** $\mathrm{FI}^{util}$
11: **end procedure**

---

## 4    FI FOR ACTIVE LEARNING.

In active learning, the primary objective is to identify the most uncertain or informative samples from an unlabeled pool for annotation, typically in sequential batches. After each round of annotation, the newly labeled data are combined with the existing labeled set to retrain the model and improve its performance. Li et al. (2024) argue that if a small perturbation to the model parameters leads to significant changes in the prediction for a given sample, this indicates high uncertainty for that sample under the current model, suggesting that it should be labeled and added to the training set. Our approach is more fundamental: since the model parameters are derived from the training data, we directly assess how perturbations to the training samples influence the model's predictions for the unlabeled samples. If slight perturbations to most training samples substantially alter the model's

prediction for a given sample, it likely contains missing information from the training set and should therefore be included.

We now introduce a method for active learning using the proposed FI. For an unlabeled sample $\boldsymbol{x}_{unlabel}$, we first assign it a predicted label and treat it as a validation sample. Next, we calculate the influence measure of $\boldsymbol{x}_{unlabel}$ with respect to each point in the training set. The overall influence measure $\text{FI}^{active}(\boldsymbol{x}_{unlabel})$ is derived by aggregating these individual measures either by averaging or using specific quantiles of these FI values. The $\text{FI}^{active}(\boldsymbol{x}_{unlabel})$ is defined as follows:

$$\text{FI}^{active}(\boldsymbol{x}_{unlabel}) = g(\{\text{FI}(\boldsymbol{z}_{unlabel}, \boldsymbol{z}_i)\}_{i=1}^n), \tag{4.1}$$

where $g$ represents the aggregation function, $\boldsymbol{z}_{unlabel} = (\boldsymbol{x}_{unlabel}, y_{pred})$, and $y_{pred}$ is the predicted label assigned by the current model for $\boldsymbol{x}_{unlabel}$.

During each round of active learning, we begin by applying Algorithm 2 to compute $\text{FI}^{active}$. We then sort all the samples in the unlabeled pool in descending order based on their $\text{FI}^{active}$ values. The top-ranked samples are labeled and added to the training set. In the subsequent round, the model is retrained, $\text{FI}^{active}$ is recalculated, and the process is repeated. For the detailed active learning algorithm, please refer to Algorithm 4.

---

**Algorithm 2** Calculation of $\text{FI}^{active}$

---

**Input:** Labeled pool of training data $\mathcal{L}$, Unlabeled pool of training data $\mathcal{U}$, Base model $\mathcal{F}$, Aggregation function $g$
**Output:** Influence measure vector $\text{FI}^{active} \in \mathbb{R}^{|\mathcal{U}| \times 1}$
1: **procedure** $\text{FI}^{active}$-CALCULATION($\mathcal{L}, \mathcal{U}, \mathcal{F}, g$)
2:     Train $\mathcal{F}$ with $\mathcal{L}$, and obtain the parameter vector $\hat{\boldsymbol{\theta}}$
3:     Generate an empty vector $\text{FI}^{active}$ of size $|\mathcal{U}| \times 1$
4:     Calculate $\boldsymbol{H}_{\hat{\boldsymbol{\theta}}}$
5:     **for** every $x_i^{\mathcal{U}}$ in $\mathcal{U}$ **do**
6:         Obtain an estimated label $\hat{y}_i$ with $\mathcal{F}$
7:         Calculate $\partial_{\boldsymbol{\theta}} L(\hat{y}_i, \boldsymbol{x}_i^{\mathcal{U}}; \hat{\boldsymbol{\theta}})$
8:         $\mathcal{J} \leftarrow \emptyset$
9:         **for** every $z_j^{\mathcal{L}}$ in $\mathcal{L}$ **do**
10:             Calculate $\boldsymbol{G}_{\boldsymbol{z}_j^{\mathcal{L}}}(\boldsymbol{\omega}_0)$ and $\partial_{\boldsymbol{x}}\partial_{\boldsymbol{\theta}} L(y_j^{\mathcal{L}}, \boldsymbol{x}_j^{\mathcal{L}}; \hat{\boldsymbol{\theta}})$
11:             $\partial_{\boldsymbol{\omega}} L(\hat{y}_i, \boldsymbol{x}_i^{\mathcal{U}}; \hat{\boldsymbol{\theta}}) \leftarrow \partial_{\boldsymbol{\theta}} L(\hat{y}_i, \boldsymbol{x}_i^{\mathcal{U}}; \hat{\boldsymbol{\theta}}) \boldsymbol{H}_{\hat{\boldsymbol{\theta}}}^{-1} \partial_{\boldsymbol{x}}\partial_{\boldsymbol{\theta}} L(y_j^{\mathcal{L}}, \boldsymbol{x}_j^{\mathcal{L}}; \hat{\boldsymbol{\theta}})$
12:             $\mathcal{J} \leftarrow \mathcal{J} \bigcup \{\partial_{\boldsymbol{\omega}} L(\hat{y}_i, \boldsymbol{x}_i^{\mathcal{U}}; \hat{\boldsymbol{\theta}}) \boldsymbol{G}_{\boldsymbol{z}_j^{\mathcal{L}}}^{\dagger}(\boldsymbol{\omega}_0) \partial_{\boldsymbol{\omega}}^{\top} L(\hat{y}_i, \boldsymbol{x}_i^{\mathcal{U}}; \hat{\boldsymbol{\theta}})\}$
13:         **end for**
14:         $\text{FI}^{active}[i] \leftarrow g(\mathcal{J})$
15:     **end for**
16:     **return** $\text{FI}^{active}$
17: **end procedure**

---

## 5 APPROXIMATION METHODS

The data selection processes described in the previous two sections have two main drawbacks. First, computing FI requires substantial storage for second-order derivatives, particularly in models with numerous parameters or high-dimensional data. To address this issue, we propose the KFSVD approximation method, which combines the Kronecker-factored (K-FAC) approximation (Martens & Grosse, 2015; Nickl et al., 2023) and Truncated Singular Value Decomposition (Truncated-SVD) approximation (Golub & Reinsch, 1971) to effectively reduce storage requirements. Sencond, the necessity to compute the $\text{FI}^{util}$ for all data points in the data trimming algorithm, as well as the need to recalculate $\text{FI}^{active}$ for all unlabeled data in each round of the active learning algorithm, significantly decreases computational efficiency. To mitigate this, we implement a subsampling approximation that enhances overall performance. The details of these two approximation methods are as follows.

**KFSVD approximation**, which combines K-FAC approximation with Truncated-SVD approximation to mitigate memory consumption associated with the Hessian matrix $\boldsymbol{H}_{\hat{\boldsymbol{\theta}}}$ and the second-order

partial derivatives $\partial_{\boldsymbol{x}}\partial_{\boldsymbol{\theta}}L(y,\boldsymbol{x};\hat{\boldsymbol{\theta}}(\boldsymbol{x}))$. The K-FAC algorithm is a widely recognized technique for approximating the Hessian matrix, which not only accelerates computations but also significantly reduces storage requirements. Meanwhile, the Truncated-SVD approximation employs power iteration and related techniques to compute the top-$k$ eigenvalues and their corresponding eigenvectors of $\partial_{\boldsymbol{x}}\partial_{\boldsymbol{\theta}}L(y,\boldsymbol{x};\hat{\boldsymbol{\theta}}(\boldsymbol{x}))$, thereby providing an effective approximation of these second-order partial derivatives. Assuming the dimensionality of the sample covariates is $d$ and the number of model parameters is $p$, the Truncated-SVD approximation enables the decomposition of the second-order partial derivatives as $\partial_{\boldsymbol{x}}\partial_{\boldsymbol{\theta}}L(y,\boldsymbol{x};\hat{\boldsymbol{\theta}}(\boldsymbol{x})) = U_{p\times k}\Lambda_{k\times k}V_{d\times k}^{\top}$, where $\Lambda_{k\times k}$ is a diagonal matrix containing the top-$k$ eigenvalues of $\partial_{\boldsymbol{x}}\partial_{\boldsymbol{\theta}}L(y,\boldsymbol{x};\hat{\boldsymbol{\theta}}(\boldsymbol{x}))$ and $U_{p\times k}$ and $V_{d\times k}$ are comprised of $k$ orthogonal vectors. Algorithm 5 and 7 provide the detailed procedures for calculating $\mathrm{FI}^{util}$ and $\mathrm{FI}^{active}$ using the KFSVD approximation.

**Subsampling approximation**, which utilizes subsampling and random forest techniques to enhance computational efficiency. Specifically, we extract a small subset of samples (e.g., 20%) and compute their FI using Algorithm 5 or 7. These computed values are then used to sort the samples. The features of each sample, along with their corresponding ranks, serve as covariates and target variables to create a new dataset. Subsequently, we train a regression model using random forests on this dataset and leverage the trained model to predict the ranks of other samples. Selections are based on these predicted ranks. Since the computation of FI is the most time-consuming part of the workflow, the overall acceleration is directly correlated with the proportion of samples for which we choose to compute FI accurately. For instance, selecting 20% of the samples can reduce the total processing time to $\frac{1}{5}$ of the original duration when the dataset is sufficiently large.

In practice, we integrate both approximation methods to develop the comprehensive $\mathrm{FI}^{util}$-based data trimming algorithm and $\mathrm{FI}^{active}$-based active learning algorithm. These algorithms effectively address the storage and computational efficiency challenges inherent in calculating FI. The detailed procedures are outlined in Algorithm 6 and 8.

To better illustrate the computational advantages of the proposed two approximation methods, we analyze their complexity in terms of both memory usage and computational speed. This analysis is also compared with the complexity of *Influence Value* (IV) as introduced by Chhabra et al. (2024).

**Time Complexity**. The following are the time complexities of FI and IV:

- $\mathrm{FI}^{util}$ without approximation methods: $\mathcal{O}(p^3 + nd^3 + ndp)$. Here, $n$ is the maximum value of the sample sizes of the training set and the validation set, $d$ is the dimension of the covariates, and $p$ is the dimension of the model parameters. The term $\mathcal{O}(p^3)$ comes from computing the inverse of the Hessian matrix. The $\mathcal{O}(nd^3)$ term is due to inverting $G$, repeated $n$ times. The $\mathcal{O}(ndp)$ term corresponds to computing $\partial_{\boldsymbol{x}}\partial_{\boldsymbol{\theta}}L$ for $n$ times.

- $\mathrm{FI}^{util}$ with approximation methods: $\mathcal{O}(\alpha ndT\log(\alpha n) + \alpha n(p+d)ks + \alpha nd^3 + \sum_{i=1}^{L} p_i^3)$. Here, $\alpha \in (0,1]$ is the proportion of data for which we compute FI accurately during subsampling. $T$ is the number of decision trees in the random forest, $k$ is the Truncated-SVD parameter, $s$ is the number of iterations for computing each eigenvalue during power iteration, and $\{p_1, p_2, \ldots, p_L\}$ represent the number of parameters in each layer of an $L$-layer neural network.

- $\mathrm{FI}^{active}$ without approximation methods: $\mathcal{O}(p^3 + n^2 d^3 + n^2 dp)$. Here, $n$ is the maximum value of the sample sizes of the labeled dataset and the unlabeled dataset.

- $\mathrm{FI}^{active}$ with approximation methods: $\mathcal{O}(\alpha ndT\log(\alpha n) + \alpha n^2(p+d)ks + \alpha n^2 d^3 + \sum_{i=1}^{L} p_i^3)$.

- IV: $\mathcal{O}(p^3 + np)$.

By employing two approximation techniques, we can significantly reduce the computational gap between our method and compared methods like IV. When the dataset is large enough, setting $\alpha n$ as a constant allows us to achieve better computational complexity than IV. For complex models (e.g., neural networks) where $p$ is sufficiently large, the $p^3$ term dominates the computational complexity of IV. In such cases, our method's computational costs could be lower than IV.

**Memory Usage**. Below is a discussion of the space complexities associated with FI and IV:

- In the computation of FI, the highest storage requirements are for the Hessian matrix and $\partial_{\boldsymbol{x}}\partial_{\boldsymbol{\theta}}L$, which together occupy a space of $\mathcal{O}(p^2 + dp)$.

- After using the KFSVD approximation method, the space requirement of FI is reduced to $\mathcal{O}(\sum_{i=1}^{L} p_i^2 + (p+d)k)$.

- In the computation of IV, it is necessary to store sample information and the Hessian matrix, resulting in a space complexity of $\mathcal{O}(p^2 + d)$.

After applying approximation techniques, our algorithm achieves a space complexity similar to that of the IV. In fact, when $p$ is large, our method offers an advantage.

## 6 EXPERIMENTAL RESULTS

In this section, we present experimental results that illustrate how our newly proposed metrics, $FI^{util}$ and $FI^{active}$, contribute to enhancing model prediction performance. We compare our algorithms with state-of-the-art strategies in both data trimming and active learning scenarios, thereby validating the effectiveness of our approach on both simulated and real-world datasets.

### 6.1 DATA TRIMMING

In this subsection, we conduct simulations using both linear and nonlinear models to demonstrate how our algorithm enhances data trimming efficiency and evaluate the effectiveness of the proposed FI on real-world datasets. The latest data trimming method, IV, serves as the primary baseline for comparison in these experiments.

**Validation on 2D Linear Model.** Logistic regression is utilized for this binary classification task. We begin by generating several datasets by sampling from two isotropic 2D Gaussian distributions. Each dataset comprises 150 training samples, 100 validation samples, and 600 test samples. The experimental settings for this

Table 1: **Comparison of two methods on linear model.** Number of cases where FI outperforms IV across 30 random seeds, along with performance improvements. *Acc_FI*: the mean accuracy by FI, and *Acc_IV*: by IV.

| # of deleted points | # of better case | Acc_FI(%) | Acc_IV(%) |
|---|---|---|---|
| 5 | 23 | 96.22±0.65 | 95.77±0.76 |
| 10 | 28 | 96.20±0.65 | 94.84±1.07 |
| 20 | 30 | 96.23±0.64 | 93.36±2.21 |

scenario are consistent with those of the study by Chhabra et al. (2024). To account for the randomness inherent in the sampling process, we analyze our method on datasets generated under the same distribution but with different random seeds. As shown in Table 1, our method consistently enhances model performance compared to theirs in most cases, particularly when trimming 5, 10, and 20 samples, with thirty different random seeds employed each time. Moreover, it is evident from Figure 1.C that IV tends to trim samples from a specific class under certain conditions. In this context, Figure 1.D clearly demonstrates that in some scenarios IV fails, while our method continues to perform effectively.

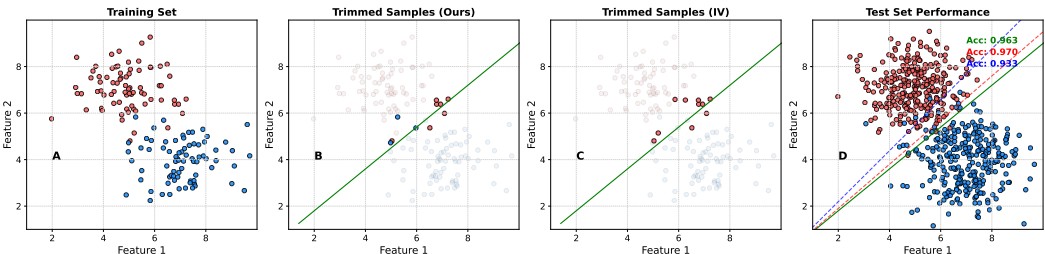

**Figure 1: Performance under Linear Model.** Different colored points represent different classes. **A** shows the training set. **B** and **C** respectively denote the samples to be trimmed by FI and IV. **D** denotes test set. **Green** line: boundary without trimming; **Red** line: boundary after FI trimming; and **Blue** line: boundary after IV trimming.

**Validation on 2D Nonlinear Model.** After demonstrating the effectiveness of our method in linear scenarios, we now extend our examination to nonlinear cases. To achieve this, we construct a binary classification dataset that is non-linearly separable, with each class represented by a crescent-shaped region. A more intuitive understanding can be obtained from Figure 9. We employ a neural network that includes an input layer, two hidden layers with ReLU activation functions, and an output layer with a sigmoid activation function. Similar to the linear case, we conduct repeated experiments in the nonlinear scenario. Each dataset comprises 500 training samples, 250 validation samples, and 250 test samples. As shown in Table 2, our method achieves a higher average accuracy and outperforms the other methods in most cases across the 20 repetitions. Figure 9 illustrates instances where utilizing IV to remove training points can lead to a deterioration in model performance under certain conditions.

**Validation on the Real-World Datasets.** In this study, we assess the efficacy of our method by utilizing four real-world datasets: two tabular datasets, *Adult* (Kohavi, 1996) and *Bank* (Moro et al., 2014); a visual dataset, *CelebA* (Liu et al., 2015); and a textual dataset, *Jigsaw Toxicity* (Noever, 2018). Additional details regarding the datasets and experiments can be found in Appendices C & D.1. Both $FI^{util}$ and IV are evaluated on the validation set, with Logistic Regression serving as the base model. The results on the test sets of these datasets are presented in Figure 2. Ablation experiments are also conducted in the Appendix E.1 to assess the impact of perturbations on model performance.

Table 2: **Comparison of two methods on nonlinear model.** Number of cases where FI outperform IV across 20 random seeds, along with performance improvements. *Acc_FI*: the mean accuracy by FI, and *Acc_IV*: by IV.

| # of deleted points | # of better case | Acc_FI(%) | Acc_IV(%) |
|---|---|---|---|
| 5 | 17 | 89.90±2.16 | 87.50±2.46 |
| 10 | 17 | 90.24±2.05 | 88.02±2.62 |
| 20 | 15 | 90.32±1.67 | 87.80±2.54 |

As illustrated in Figure 2, our findings indicate that under a limited budget $b$, our $FI^{util}$ data trimming method consistently outperforms two baseline models, IV and Random Trimming. Among the four datasets examined, Random Trimming demonstrates no improvement in model performance. Although IV exhibits a notable enhancement on *Adult* and *Bank*, it tends to remove important data points on *CelebA*, resulting in decreased performance, and shows no improvement on *Jigsaw Toxicity* compared to Random Trimming. This suggests that IV may fail in certain scenarios. In contrast, our method consistently achieves the maximum improvements across all datasets, particularly on the *Bank* dataset, where accuracy increases by more than 10%. Even on *CelebA*, where IV fails completely, our model still enhances the test set accuracy, further demonstrating the robustness of our approach. To ensure a more rigorous comparison between FI and IV, we also perform K-fold cross-validation in Appendix E.5. The results demonstrate that our method consistently outperforms IV, highlighting the robustness and stability of our approach.

The experimental results above show that our method performs better on real-world datasets than on simulated ones. This is largely due to the simplicity of the simulated datasets, which are two-dimensional with clear boundaries effectively separating the classes and contain no erroneous data points. In such cases, a sufficiently large dataset allows the model to easily find the optimal boundary, minimizing the advantages of data trimming. In contrast, real-world datasets are typically high-dimensional and more complex, often containing errors. Here, the benefits of removing potentially erroneous high-influence points become more evident. To support this, we introduce noise into the real-world datasets and conduct further experiments; the details of the noise addition method are given in Appendix D.1, and the results are shown in Figure 8. The findings clearly demonstrate that both FI and IV outperform Random Trimming, with our method retaining significant advantages over IV, further confirming its effectiveness.

Finally, although the Shapley-value-based method is time-consuming, we conduct a comparison with our approach using specific datasets. And we also examine the potential impact of the masking effect. Details of both analyses are presented in Appendix E.4.

## 6.2 ACTIVE LEARNING

In this section, we primarily conduct numerical experiments on three simple tabular datasets and three complex image datasets to demonstrate the effectiveness of the proposed FI metric in active learning. We compare our method with two state-of-the-art active learning baselines including IV

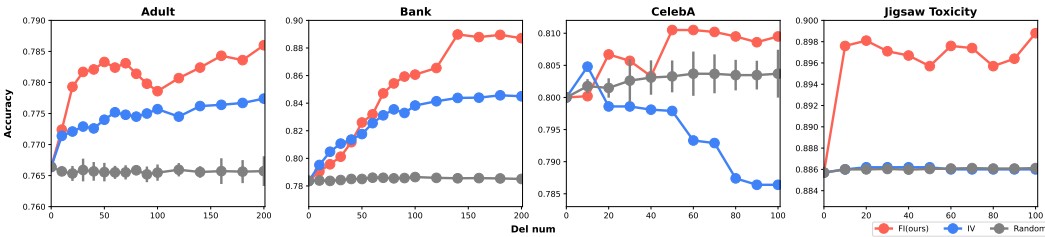

**Figure 2:** Accuracy curves of three data trimming methods on test sets of *Adult*, *Bank*, *CelebA* and *Jigsaw Toxicity*.

(Chhabra et al., 2024) and BALD (Gal et al., 2017; Kirsch et al., 2019; 2023). Random Selection is also included as a baseline. All results are averaged over three runs to ensure a reliable evaluation.

Note that, IV, as proposed by Chhabra et al. (2024), originally calculates influence values only once during the initial selection, which can hinder overall performance since these values are not updated with subsequent labelings. To ensure fairness in our comparisons, we modified IV to recalculate influence values in each round of selection, and this updated version is used in our experiments.

For our method, the aggregation function $g(\{\mathrm{FI}(z_{unlabel}, z_i)\}_{i=1}^n)$ in Equation 4.1 denotes the operation of calculating the mean of the data points from the set $\{\mathrm{FI}(z_{unlabel}, z_i)\}_{i=1}^n$ that fall between the 10th and 90th percentiles, sorted in descending order. In other words, we compute the mean of the middle 80% of the data after sorting. This approach effectively excludes extreme values, thereby enhancing the robustness of our final result.

**Tabular Datasets.** For the tabular datasets, we employed a Logistic Regression model. The total number of annotation rounds for each experiment, along with the settings for the unlabeled pool size and acquisition size, are detailed in Table 4. As shown in Figure 3, our method outperforms the other approaches on these simple tabular datasets. Random Selection performs reasonably well initially, but as data volume increases, improvements in model performance diminish. BALD shows comparable performance to Random Selection on tabular datasets. IV initially underperforms compared to Random Selection, but as more data is added, its performance improves and eventually becomes comparable to BALD. Our method starts similarly to BALD, but its focus on more challenging samples allows it to quickly address missing information once the model achieves a certain accuracy. As a result, in the latter stages of each graph, our method distinctly diverges from the others.

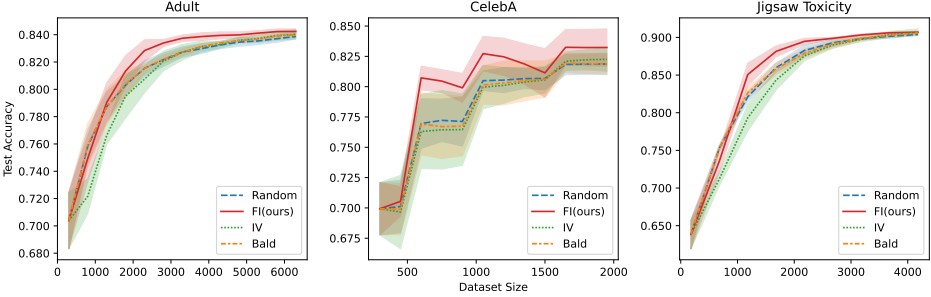

**Figure 3:** Classification performance due to the different active learning methods on *Adult*, *CelebA*, and *Jigsaw Toxicity*.

**Image Classification.** The data in the tabular datasets have been processed, resulting in high test accuracy with logistic regression. In contrast, for the image datasets—*MNIST* (Lecun et al., 1998), *EMNIST* (Cohen et al., 2017), and *CIFAR-10* (Krizhevsky, 2012)—the input consists of raw images, and we employ a Convolutional Neural Network (CNN) as the classifier. To simplify the computation of $\mathrm{FI}^{active}$, we utilize the outputs from the last layer of the neural network and focus on the parameters of that layer. However, during each training iteration, all parameters of the neural network are updated,

not just those of the final layer. The settings for the unlabeled pool size and acquisition size for each image dataset are provided in Table 5 . Experimental results indicate that our method maintains comparable time consumption to other approaches, even with complex image data (see Table 9 for details). Our primary focus in active learning is enhancing model accuracy rather than speed. We developed CNNs tailored for *MNIST* and *EMNIST*, with specifications in Table 6. For *CIFAR-10*, we adopted a model architecture from Trockman & Kolter (2022). As shown in Figure 4, both BALD and IV methods are suited to different scenarios, but our method consistently outperforms others across all datasets, particularly in more complex situations. In fact, the *MNIST* dataset, being relatively straightforward, does not fully showcase the advantages of our proposed method. To address this limitation, we constructed more challenging variants of the *MNIST* dataset, specifically unbalanced and redundant versions. Furthermore, we evaluated our method on two more complex datasets, *Office-31* and *AG News*, and experimented with fine-tuning pre-trained models. Comparative analyses conducted on these datasets demonstrate the superior efficacy of our method, as detailed in Appendix E.3. Additionally, Figure 5 visually represents the selection preferences of $FI^{active}$, highlighting its tendency to identify points that are more challenging for the current model to distinguish.

To deepen our analysis, we present ablation experiments in Appendix E.1 to better understand the trade-offs between potential performance reduction and quantified computational cost reduction, and we also conduct a comparison with the Shapley-value-based method in Appendix E.4.

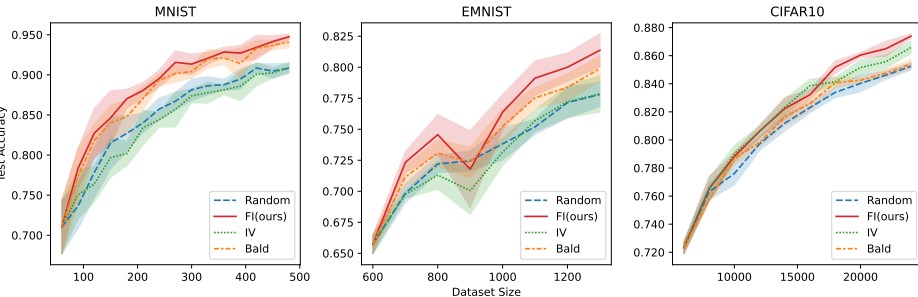

**Figure 4:** Classification performance due to the different active learning methods on *MNIST*, *EMNIST*, and *CIFAR-10*.

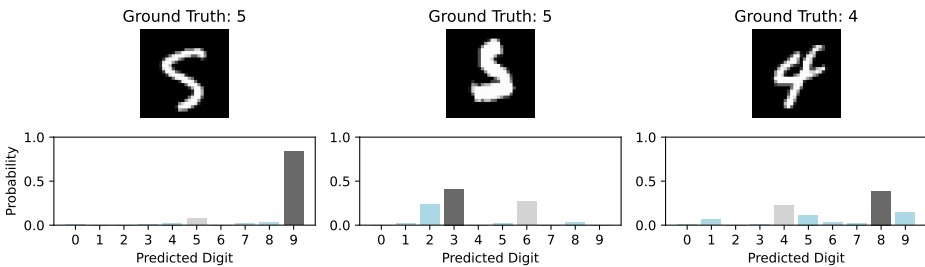

**Figure 5:** Images with the highest $FI^{active}$ selected in the first round of active learning for the *MNIST* problem, along with their corresponding prediction probability distributions.

## 7  CONCLUSION

In this paper, we introduce a novel local influence metric that evaluates the impact of perturbations to training samples on model performance concerning validation samples. This metric is applicable in both data trimming and active learning, offering valuable insights into the contributions of individual samples and their relationships with unlabeled data. Furthermore, we propose two approximation methods to mitigate the computational costs associated with calculating local influence measures. Our experimental results demonstrate that these algorithms effectively reduce costs while outperforming other state-of-the-art methods.

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

## A  DERIVATION OF EXPRESSION FOR $\partial_{\boldsymbol{\omega}} \hat{\boldsymbol{\theta}}(\boldsymbol{x} + \boldsymbol{\omega})\big|_{\boldsymbol{\omega}=\boldsymbol{\omega}_0}$

Recall that $\hat{\boldsymbol{\theta}}(\boldsymbol{x})$ minimizes the empirical risk: $R(\boldsymbol{\theta}) := \frac{1}{n} \sum_{i=1}^{n} L(y_i, \boldsymbol{x}_i; \boldsymbol{\theta})$. We assume that $R$ is twice-differentiable and strongly convex in $\boldsymbol{\theta}$, i.e., $\boldsymbol{H}_{\hat{\boldsymbol{\theta}}} := \partial_{\boldsymbol{\theta}}^2 R(\hat{\boldsymbol{\theta}}) = \frac{1}{n} \sum_{i=1}^{n} \partial_{\boldsymbol{\theta}}^2 L(y_i, \boldsymbol{x}_i; \hat{\boldsymbol{\theta}})$ exists and is positive definite. This ensures the availability of $\boldsymbol{H}_{\hat{\boldsymbol{\theta}}}^{-1}$, which will be utilized in the subsequent derivation. The perturbed parameter vector $\hat{\boldsymbol{\theta}}(\boldsymbol{x} + \boldsymbol{\omega})$ can be written as

$$\hat{\boldsymbol{\theta}}(\boldsymbol{x} + \boldsymbol{\omega}) = \arg\min_{\boldsymbol{\theta} \in \Theta} \{ R(\boldsymbol{\theta}) + \frac{1}{n} L(y, \boldsymbol{x} + \boldsymbol{\omega}; \boldsymbol{\theta}) - \frac{1}{n} L(y, \boldsymbol{x}; \boldsymbol{\theta}) \}. \tag{A.1}$$

Define the parameter change $\delta(\boldsymbol{\omega}) = \hat{\boldsymbol{\theta}}(\boldsymbol{x} + \boldsymbol{\omega}) - \hat{\boldsymbol{\theta}}(\boldsymbol{x})$, and note that, since $\hat{\boldsymbol{\theta}}(\boldsymbol{x})$ does not depend on $\boldsymbol{\omega}$, the quantity we aim to calculate can be expressed in terms of it: $\partial_{\boldsymbol{\omega}} \hat{\boldsymbol{\theta}}(\boldsymbol{x} + \boldsymbol{\omega}) = \partial_{\boldsymbol{\omega}} \delta(\boldsymbol{\omega})$.

According to the definition of $\hat{\boldsymbol{\theta}}(\boldsymbol{x} + \boldsymbol{\omega})$, we know

$$\boldsymbol{0} = \partial_{\boldsymbol{\theta}} R(\hat{\boldsymbol{\theta}}(\boldsymbol{x} + \boldsymbol{\omega})) + \frac{1}{n}\partial_{\boldsymbol{\theta}} L(y, \boldsymbol{x} + \boldsymbol{\omega}; \hat{\boldsymbol{\theta}}(\boldsymbol{x} + \boldsymbol{\omega})) - \frac{1}{n}\partial_{\boldsymbol{\theta}} L(y, \boldsymbol{x}; \hat{\boldsymbol{\theta}}(\boldsymbol{x} + \boldsymbol{\omega})). \tag{A.2}$$

Next, since $\hat{\boldsymbol{\theta}}(\boldsymbol{x} + \boldsymbol{\omega}) \to \hat{\boldsymbol{\theta}}(\boldsymbol{x})$ as $\boldsymbol{\omega} \to \boldsymbol{0}$, we perform a Taylor expansion of the right-hand side:

$$\boldsymbol{0} = \partial_{\boldsymbol{\theta}} R(\hat{\boldsymbol{\theta}}(\boldsymbol{x} + \boldsymbol{\omega})) + \frac{1}{n}\partial_{\boldsymbol{\theta}} L(y, \boldsymbol{x} + \boldsymbol{\omega}; \hat{\boldsymbol{\theta}}(\boldsymbol{x} + \boldsymbol{\omega})) - \frac{1}{n}\partial_{\boldsymbol{\theta}} L(y, \boldsymbol{x}; \hat{\boldsymbol{\theta}}(\boldsymbol{x} + \boldsymbol{\omega})), \tag{A.3}$$

$$\approx [\partial_{\boldsymbol{\theta}} R(\hat{\boldsymbol{\theta}}(\boldsymbol{x})) + \frac{1}{n}\partial_{\boldsymbol{\theta}} L(y, \boldsymbol{x} + \boldsymbol{\omega}; \hat{\boldsymbol{\theta}}(\boldsymbol{x})) - \frac{1}{n}\partial_{\boldsymbol{\theta}} L(y, \boldsymbol{x}; \hat{\boldsymbol{\theta}}(\boldsymbol{x}))]$$

$$+ [\partial_{\boldsymbol{\theta}}^2 R(\hat{\boldsymbol{\theta}}(\boldsymbol{x})) + \frac{1}{n}\partial_{\boldsymbol{\theta}}^2 L(y, \boldsymbol{x} + \boldsymbol{\omega}; \hat{\boldsymbol{\theta}}(\boldsymbol{x})) - \frac{1}{n}\partial_{\boldsymbol{\theta}}^2 L(y, \boldsymbol{x}; \hat{\boldsymbol{\theta}}(\boldsymbol{x}))]\delta(\boldsymbol{\omega}). \tag{A.4}$$

Solving for $\delta(\boldsymbol{\omega})$, we get

$$\delta(\boldsymbol{\omega}) \approx - [\partial_{\boldsymbol{\theta}}^2 R(\hat{\boldsymbol{\theta}}(\boldsymbol{x})) + \frac{1}{n}\partial_{\boldsymbol{\theta}}^2 L(y, \boldsymbol{x} + \boldsymbol{\omega}; \hat{\boldsymbol{\theta}}(\boldsymbol{x})) - \frac{1}{n}\partial_{\boldsymbol{\theta}}^2 L(y, \boldsymbol{x}; \hat{\boldsymbol{\theta}}(\boldsymbol{x}))]^{-1}$$

$$\cdot [\partial_{\boldsymbol{\theta}} R(\hat{\boldsymbol{\theta}}(\boldsymbol{x})) + \frac{1}{n}\partial_{\boldsymbol{\theta}} L(y, \boldsymbol{x} + \boldsymbol{\omega}; \hat{\boldsymbol{\theta}}(\boldsymbol{x})) - \frac{1}{n}\partial_{\boldsymbol{\theta}} L(y, \boldsymbol{x}; \hat{\boldsymbol{\theta}}(\boldsymbol{x}))]. \tag{A.5}$$

Since $\hat{\boldsymbol{\theta}}(\boldsymbol{x})$ minimizes $R$, we have $\partial_{\boldsymbol{\theta}} R(\hat{\boldsymbol{\theta}}) = 0$. We further assume that $\partial_{\boldsymbol{\theta}}^2 L(y, \boldsymbol{x}; \hat{\boldsymbol{\theta}}(\boldsymbol{x}))$ is continuous on $\boldsymbol{x}$, then we have

$$\delta(\boldsymbol{\omega}) \approx -\frac{1}{n}[\partial_{\boldsymbol{\theta}}^2 R(\hat{\boldsymbol{\theta}}(\boldsymbol{x}))]^{-1}[\frac{1}{n}\partial_{\boldsymbol{\theta}} L(y, \boldsymbol{x} + \boldsymbol{\omega}; \hat{\boldsymbol{\theta}}(\boldsymbol{x})) - \frac{1}{n}\partial_{\boldsymbol{\theta}} L(y, \boldsymbol{x}; \hat{\boldsymbol{\theta}}(\boldsymbol{x}))]. \tag{A.6}$$

After differentiation, the final expression can be obtained,

$$\partial_{\boldsymbol{\omega}}\hat{\boldsymbol{\theta}}(\boldsymbol{x} + \boldsymbol{\omega})\big|_{\boldsymbol{\omega}=\boldsymbol{\omega}_0} = \partial_{\boldsymbol{\omega}}\delta(\boldsymbol{\omega})\big|_{\boldsymbol{\omega}=\boldsymbol{\omega}_0} \approx -\frac{1}{n}\boldsymbol{H}_{\hat{\boldsymbol{\theta}}}^{-1}\partial_{\boldsymbol{x}}\partial_{\boldsymbol{\theta}} L(y, \boldsymbol{x}; \hat{\boldsymbol{\theta}}(\boldsymbol{x})). \tag{A.7}$$

# B  ALGORITHMS

---

**Algorithm 3** Data Trimming

---

**Input:** Training set $\mathcal{Z}$, Validation set $\mathcal{V}$, Base model $\mathcal{F}$, Budget $b$
**Output:** Trimmed Dataset $\mathcal{Z}'$

1: **procedure** DATATRIMMING($\mathcal{Z}, \mathcal{V}, \mathcal{F}, b$)
2:     Call the algorithm FI$^{util}$-CALCULATION($\mathcal{Z}, \mathcal{V}, \mathcal{F}$) to obtain FI$^{util} \in \mathbb{R}^{|\mathcal{Z}|\times 1}$
3:     Select $b$ samples $\boldsymbol{z}_i \in \mathcal{Z}$ as $\{Z_b\}$, whose FI$^{util}[i]$ rank in the top $b$
4:     $\mathcal{Z}' \leftarrow \mathcal{Z} \setminus \{Z_b\}$
5:     **return** $\mathcal{Z}'$
6: **end procedure**

---

**Algorithm 4** Active Learning

---

**Input:** Labeled pool of training data $\mathcal{L}$, Unlabeled pool of training data $\mathcal{U}$, Base model $\mathcal{F}$, The number of samples for annotation (per round) $N$, Aggregation function $g$
**Output:** Updated labeled pool $\mathcal{L}$

1: **procedure** ACTIVELEARNING($\mathcal{L}, \mathcal{U}, \mathcal{F}, N, g$)
2:     **for** $j \leftarrow 1$ to $NUM\_rounds$ **do**
3:         Call the algorithm FI$^{active}$-CALCULATION($\mathcal{L}, \mathcal{U}, \mathcal{F}, g$) to obtain FI$^{active} \in \mathbb{R}^{|\mathcal{U}|\times 1}$
4:         Select $N$ samples $x_i^{\mathcal{U}}$ as $\{X_N\}$, whose FI$^{active}[i]$ ranks in the top $N$
5:         Take $\{X_N\}$ out of $\mathcal{U}$, and query their labels $\{Y_N\}$
6:         Update $\mathcal{L} \leftarrow \mathcal{L} \bigcup \{X_N, Y_N\}$
7:     **end for**
8:     **return** $\mathcal{L}$
9: **end procedure**

---

---

**Algorithm 5** Calculation of $\text{FI}_{approx}^{util}$

---

**Input:** Training set $\mathcal{Z}$, Validation set $\mathcal{V}$, Base model $\mathcal{F}$, Truncated-SVD parameter $k$
**Output:** Influence measure vector $\text{FI}_{approx}^{util} \in \mathbb{R}^{|\mathcal{Z}| \times 1}$

1: **procedure** $\text{FI}_{approx}^{util}$-CALCULATION($\mathcal{Z}$, $\mathcal{V}$, $\mathcal{F}$, $k$)
2:     Train $\mathcal{F}$ with $\mathcal{Z}$, and obtain the parameter vector $\hat{\boldsymbol{\theta}}$
3:     Generate an empty vector $\text{FI}_{approx}^{util}$ of size $|\mathcal{Z}| \times 1$
4:     Calculate $\boldsymbol{H}_{\hat{\boldsymbol{\theta}}}$ with K-FAC approximation
5:     Calculate $\frac{1}{|\mathcal{V}|} \sum_{(\boldsymbol{x}', y') \in \mathcal{V}} \partial_{\boldsymbol{\theta}} L(y', \boldsymbol{x}'; \hat{\boldsymbol{\theta}})$
6:     **for** every $z_i$ in $\mathcal{Z}$ **do**
7:         Calculate $\boldsymbol{G}_{\boldsymbol{z}_i}(\boldsymbol{\omega}_0)$
8:         Calculate $\Lambda_{k \times k}$, $U_{p \times k}$ and $V_{d \times k}$ with power iteration
9:         $\partial_{\boldsymbol{x}} \partial_{\boldsymbol{\theta}} L(y_i, \boldsymbol{x}_i; \hat{\boldsymbol{\theta}}) \leftarrow U_{p \times k} \Lambda_{k \times k} V_{d \times k}^{\top}$
10:         $\partial_{\boldsymbol{\omega}} L(\mathcal{V}; \hat{\boldsymbol{\theta}}) \leftarrow \left[ \frac{1}{|\mathcal{V}|} \sum_{(\boldsymbol{x}', y') \in \mathcal{V}} \partial_{\boldsymbol{\theta}} L(y', \boldsymbol{x}'; \hat{\boldsymbol{\theta}}) \right] \boldsymbol{H}_{\hat{\boldsymbol{\theta}}}^{-1} \partial_{\boldsymbol{x}} \partial_{\boldsymbol{\theta}} L(y_i, \boldsymbol{x}_i; \hat{\boldsymbol{\theta}})$
11:         $\text{FI}_{approx}^{util}[i] \leftarrow \partial_{\boldsymbol{\omega}} L(\mathcal{V}; \hat{\boldsymbol{\theta}}) \boldsymbol{G}_{\boldsymbol{z}_i}^{\dagger}(\boldsymbol{\omega}_0) \partial_{\boldsymbol{\omega}}^{\top} L(\mathcal{V}; \hat{\boldsymbol{\theta}})$
12:     **end for**
13:     **return** $\text{FI}_{approx}^{util}$
14: **end procedure**

---

**Algorithm 6** Data Trimming with Approximation

---

**Input:** Training set $\mathcal{Z}$, Validation set $\mathcal{V}$, Base model $\mathcal{F}$, Budget $b$, Truncated-SVD parameter $k$, Subsampling rate $\alpha \in (0, 1]$
**Output:** Trimmed Dataset $\mathcal{Z}'$

1: **procedure** DATATRIMMINGAPPROX($\mathcal{Z}$, $\mathcal{V}$, $\mathcal{F}$, $b$, $k$, $\alpha$)
2:     Subsample $\mathcal{Z}_{\alpha} \subseteq \mathcal{Z}$ s.t. $z \in \mathcal{Z}_{\alpha}$ w.p. $\alpha$ for all $z \in \mathcal{Z}$
3:     Call the algorithm $\text{FI}_{approx}^{util}$-CALCULATION($\mathcal{Z}_{\alpha}$, $\mathcal{V}$, $\mathcal{F}$, $k$) to obtain $\text{FI}_{approx}^{util} \in \mathbb{R}^{|\mathcal{Z}_{\alpha}| \times 1}$
4:     **for** $z_{i'} = (x_{i'}, y_{i'}) \in \mathcal{Z}_{\alpha}$ **do**
5:         $r_{i'} \leftarrow$ the rank of $\text{FI}_{approx}^{util}[i']$ sorted in ascending order
6:     **end for**
7:     Train a random forest $h$ with $\{(x_{i'}, r_{i'})\}_{i'=1}^{|\mathcal{Z}_{\alpha}|}$
8:     **for** $z_i = (x_i, y_i) \in \mathcal{Z}$ **do**
9:         $\hat{r}_i \leftarrow h(x_i)$
10:     **end for**
11:     Select $b$ samples $\boldsymbol{z}_i \in \mathcal{Z}$, whose $\hat{r}_i$ rank in the top $b$
12:     $\mathcal{Z}' \leftarrow \mathcal{Z} \setminus \{Z_b\}$
13:     **return** $\mathcal{Z}'$
14: **end procedure**

---

---

**Algorithm 7** Calculation of $\text{FI}_{approx}^{active}$

---

**Input:** Labeled pool of training data $\mathcal{L}$, Unlabeled pool of training data $\mathcal{U}$, Base model $\mathcal{F}$, Aggregation function $g$, Truncated-SVD parameter $k$
**Output:** Influence measure vector $\text{FI}_{approx}^{active} \in \mathbb{R}^{|\mathcal{U}| \times 1}$

1: **procedure** $\text{FI}_{approx}^{active}$-CALCULATION($\mathcal{L}, \mathcal{U}, \mathcal{F}, g, k$)
2:    Train $\mathcal{F}$ with $\mathcal{L}$, and obtain the parameter vector $\hat{\boldsymbol{\theta}}$
3:    Generate an empty vector $\text{FI}_{approx}^{active}$ of size $|\mathcal{U}| \times 1$
4:    Calculate $\boldsymbol{H}_{\hat{\boldsymbol{\theta}}}$ with K-FAC approximation
5:    **for** every $x_i^{\mathcal{U}}$ in $\mathcal{U}$ **do**
6:      Obtain an estimated label $\hat{y}_i$ with $\mathcal{F}$
7:      Calculate $\partial_{\boldsymbol{\theta}} L(\hat{y}_i, \boldsymbol{x}_i^{\mathcal{U}}; \hat{\boldsymbol{\theta}})$
8:      $\mathcal{J} \leftarrow \emptyset$
9:      **for** every $z_k^{\mathcal{L}}$ in $\mathcal{L}$ **do**
10:        Calculate $\boldsymbol{G}_{\boldsymbol{z}_k^{\mathcal{L}}}(\boldsymbol{\omega}_0)$
11:        Calculate $\Lambda_{k \times k}$, $U_{p \times k}$ and $V_{d \times k}$ with power iteration
12:        $\partial_{\boldsymbol{x}} \partial_{\boldsymbol{\theta}} L(y_k^{\mathcal{L}}, \boldsymbol{x}_k^{\mathcal{L}}; \hat{\boldsymbol{\theta}}) \leftarrow U_{p \times k} \Lambda_{k \times k} V_{d \times k}^{\top}$
13:        $\partial_{\boldsymbol{\omega}} L(\hat{y}_i, \boldsymbol{x}_i^{\mathcal{U}}; \hat{\boldsymbol{\theta}}) \leftarrow \partial_{\boldsymbol{\theta}} L(\hat{y}_i, \boldsymbol{x}_i^{\mathcal{U}}; \hat{\boldsymbol{\theta}}) \boldsymbol{H}_{\hat{\boldsymbol{\theta}}}^{-1} \partial_{\boldsymbol{x}} \partial_{\boldsymbol{\theta}} L(y_k^{\mathcal{L}}, \boldsymbol{x}_k^{\mathcal{L}}; \hat{\boldsymbol{\theta}})$
14:        $\mathcal{J} \leftarrow \mathcal{J} \bigcup \{\partial_{\boldsymbol{\omega}} L(\hat{y}_i, \boldsymbol{x}_i^{\mathcal{U}}; \hat{\boldsymbol{\theta}}) \boldsymbol{G}_{\boldsymbol{z}_k^{\mathcal{L}}}^{\dagger}(\boldsymbol{\omega}_0) \partial_{\boldsymbol{\omega}}^{\top} L(\hat{y}_i, \boldsymbol{x}_i^{\mathcal{U}}; \hat{\boldsymbol{\theta}})\}$
15:      **end for**
16:      $\text{FI}_{approx}^{active}[i] \leftarrow g(\mathcal{J})$
17:    **end for**
18:    **return** $\text{FI}_{approx}^{active}$
19: **end procedure**

---

---

**Algorithm 8** Active Learning with Approximation

---

**Input:** Labeled pool of training data $\mathcal{L}$, Unlabeled pool of training data $\mathcal{U}$, Base model $\mathcal{F}$, Number of samples for annotation (per round) $N$, Aggregation function $g$, Truncated-SVD parameter $k$, Subsampling rate $\alpha \in (0, 1]$
**Output:** Updated labeled pool $\mathcal{L}$

1: **procedure** ACTIVELEARNINGAPPROX($\mathcal{L}, \mathcal{U}, \mathcal{F}, N, g, k, \alpha$)
2:    **for** $j \leftarrow 1$ to $NUM\_rounds$ **do**
3:      Subsample $\mathcal{U}_{\alpha} \subseteq \mathcal{U}$ s.t. $x \in \mathcal{U}_{\alpha}$ w.p. $\alpha$ for all $x \in \mathcal{U}$
4:      Call the algorithm $\text{FI}_{approx}^{active}$-CALCULATION($\mathcal{L}, \mathcal{U}_{\alpha}, \mathcal{F}, g, k$) to obtain $\text{FI}_{approx}^{active} \in \mathbb{R}^{|\mathcal{U}_{\alpha}| \times 1}$
5:      **for** $x_{i'} \in \mathcal{U}_{\alpha}$ **do**
6:        $r_{i'} \leftarrow$ the rank of $\text{FI}_{approx}^{active}[i']$ sorted in ascending order
7:      **end for**
8:      Train a random forest $h$ with $\{(x_{i'}, r_{i'})\}_{i'=1}^{|\mathcal{U}_{\alpha}|}$
9:      **for** $x_i \in \mathcal{U}$ **do**
10:        $\hat{r}_i \leftarrow h(x_i)$
11:      **end for**
12:      Select $N$ samples $x_i$ as $\{X_N\}$, whose $\hat{r}_i$ ranks in the top $N$
13:      Take $\{X_N\}$ out of $\mathcal{U}$, and query their labels $\{Y_N\}$
14:      Update $\mathcal{L} \leftarrow \mathcal{L} \bigcup \{X_N, Y_N\}$
15:    **end for**
16:    **return** $\mathcal{L}$
17: **end procedure**

---

## C  DATA SOURCES

**Adult.** The Adult dataset consists of 48,842 instances and 14 features, including categorical and integer types. Extracted from the 1994 Census database by Barry Becker, it focuses on predicting whether an individual's annual income exceeds \$50,000. Records were filtered based on criteria such

as age, gross income, and work hours, making this dataset a valuable resource for classification tasks in social science and *Access Link*: *Adult Database*

**Bank.** The Bank dataset pertains to direct marketing campaigns conducted by a Portuguese banking institution, focusing on phone call outreach. The primary objective of this dataset is to classify whether a client will subscribe to a term deposit, indicated by the binary variable (yes/no). This dataset is derived from multiple marketing campaigns, which often required several contacts with the same client to ascertain their interest in the product. *Access Link*: *Bank Database*

**CelebA.** The CelebA dataset is a large-scale facial attribute dataset containing over 200,000 celebrity images, each annotated with 40 attribute labels. Due to its rich annotations and diversity in facial appearances, CelebA has been widely used in various computer vision tasks, such as facial recognition, attribute prediction, and generative modeling. *Access Link*: *CelebA Database*

**Jigsaw Toxicity.** The Jigsaw dataset of Wikipedia consists of comments from online platforms that have been labeled for toxicity. It contains a large number of comments, with 28 features of syntax, sentiment, emotion and outlier word dictionaries. The dataset is commonly used for training and evaluating machine learning models aimed at detecting harmful or inappropriate content in user-generated text. *Access Link*: *Jigsaw Toxicity Database*

**MNIST.** The MNIST database is a large collection of handwritten digits that is widely used for training and testing in the field of machine learning. This dataset contains 70,000 images of handwritten digits (0-9), each of which is a $28 \times 28$ pixel grayscale image. *Access Link*: *MNIST Database*

**EMNIST.** The Extended MNIST database consists of handwritten character digits sourced from the NIST Special Database 19, formatted as $28 \times 28$ pixel images to align with the structure of the MNIST dataset. There are six different splits provided in this dataset, and we use the EMNIST Letters with 145,600 characters. *Access Link*: *EMNIST Database*

**CIFAR10.** The CIFAR10 database, developed by the Canadian Institute for Advanced Research, is a widely utilized collection of images for training machine learning and computer vision algorithms. It consists of 60,000 color images, each measuring $32 \times 32$ pixels, categorized into 10 classes: airplanes, cars, birds, cats, deer, dogs, frogs, horses, ships, and trucks. Each class contains 6,000 images. *Access Link*: *CIFAR10 Database*

**Office-31.** The Office-31 dataset is a widely used benchmark dataset for domain adaptation tasks. It contains 4,110 images categorized into 31 classes, collected from three different domains: Amazon (**A**), Webcam (**W**), and DSLR (**D**). These domains represent images from an online retailer, a web camera, and a digital SLR camera, respectively. In our experiments, we combine all domains together, treating the dataset as a single unified classification task. *Access Link*: *Office-31 Dataset*

**AG News.** The AG News dataset is a widely used text classification dataset that contains news articles categorized into four distinct classes: World, Sports, Business, and Science/Technology. The dataset consists of 120,000 training samples and 7,600 test samples, with each class containing an equal number of articles. It is commonly used for benchmarking text classification models in natural language processing (NLP). *Access Link*: *AG News Dataset*

## D  EXPERIMENT DETAILS

### D.1  EXPERIMENT DETAILS IN 6.1

**Data Construction for Real-World Datasets.**    In this experiment, the preprocessing methods for all real-word datasets are consistent with those used in Chhabra et al. (2024). For *CelebA*, we use the extracted features provided by the authors Liu et al. (2015), and for *Jigsaw Toxicity*, we obtain text embeddings using the MiniLM transformer model (Wang et al., 2020). Further details are provided below.

- **Adult.** This dataset contains 37,692 samples, with 30,162 for training and 7,530 for testing. There are 102 features, and the target is to predict if income exceeds $50k (yes) or not (no).
- **Bank.** This dataset consists of 30,490 samples, divided into 18,292 training samples and 12,198 test samples. There are 50 features, and the target is to predict if the client will subscribe a term deposit (yes/no).

- **CelebA.** This dataset includes 104,163 samples, split into 62,497 training samples and 41,666 testing samples. There are 39 features, and the aim is to predict whether a person is smiling (yes) or not (no).

- **Jigsaw Toxicity.** This dataset consists of 30,000 samples, split into 18,000 training samples and 12,000 test samples. There are 385 features, and the target is to determine if a tweet is toxic (yes) or not (no).

Given that the initial test accuracy of the model on the original datasets generally exceeds 90%, the impact of data trimming is minimal. Accordingly, we randomly sample 5,000 instances from the original training set to serve as the new training set, and similarly, 4,200 instances from the original test set to serve as the new test set.

**Data Construction for Noisy Real-World Datasets.** Consider the additional experiments on data trimming presented in Figure 8. For *Adult*, *Bank*, and *Jigsaw Toxicity*, our construction method follows the same approach as in the Real-World Datasets experiment, with the additional step of randomly sampling 500 instances from the training sets and adding white noise with a variance of 0.1. For *CelebA*, since the variables are binary (-1, 1), we introduced noise by randomly selecting six feature columns and flipping their values (transforming -1 to 1 and 1 to -1). This approach generates a noisy dataset for supplementary experiments.

**Random Trimming.** We randomly remove $b$ (budget) data points from the training sets under five random seeds, and evaluate the average performance on the test sets in each iteration.

**Experimental Procedure and Parameter Settings.** First, a model is trained on the initial dataset. Based on this trained model, we then implement three data trimming strategies, removing $b$ data points. Finally, we retrain the model to evaluate the effectiveness of the different trimming strategies. The experimental procedure is illustrated in Figure 6, highlighting the key steps involved in our study. Additionally, the parameter settings are detailed in Table 3.

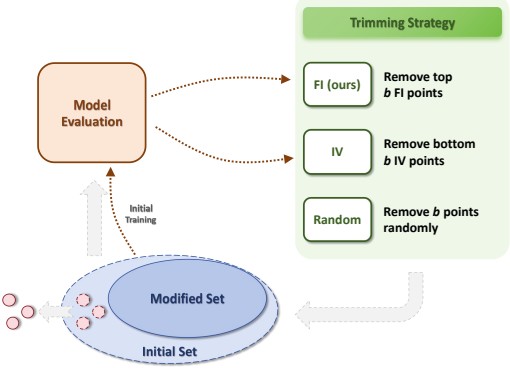

**Figure 6:** Flowchart of data trimming.

**Table 3:** Parameter Settings of Data Trimming.

| | Adult | Bank | CelebA | Jigsaw Toxicity |
|---|---|---|---|---|
| **optimizer** | SGD | SGD | Adam | Adam |
| **learning_rate** | 1e-2 | 1e-2 | 1e-4 | 1e-2 |
| **weight_decay** | 1e-2 | 1e-2 | 1e-6 | 1e-2 |

| | Adult +noise | Bank +noise | CelebA +noise | Jigsaw Toxicity +noise |
|---|---|---|---|---|
| **optimizer** | SGD | SGD | SGD | Adam |
| **learning_rate** | 1e-2 | 1e-2 | 1e-2 | 1e-1 |
| **weight_decay** | 1e-2 | 1e-4 | 1e-6 | 1e-4 |

## D.2 EXPERIMENT DETAILS IN 6.2

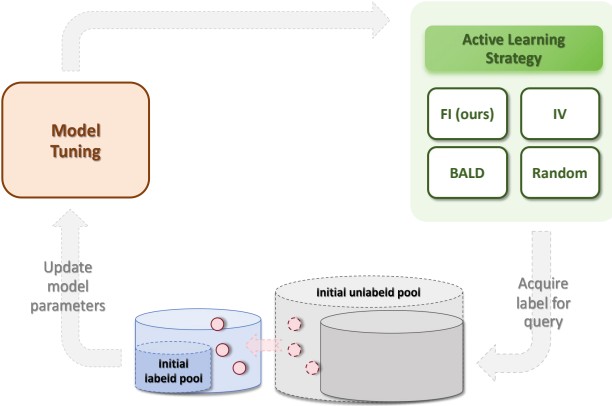

**Figure 7:** Flowchart of active learning.

**Data Construction for Unbalanced and Redundant MNIST.** For *Unbalanced MNIST*, we set the total sample size to 27,500. Categories 0 to 4 are each allocated an equal sample size, representing 1/55 of the total sample. Similarly, categories 5 to 9 are assigned equal sample sizes, with each constituting 10/55 of the total sample. This allocation strategy ensures a deliberate imbalance among the classes. Samples are systematically drawn based on these ratios from the original dataset to create this new unbalanced dataset. For *Redundant MNIST*, the task is delineated to classify solely the digits 1 and 7, presented in equal proportions. If the acquisition function selects an input from any class other than 1 or 7, the labeling function designates a "neither" category. This setup leads to a three-way classification scheme during training, categorized as 1 vs. 7 vs. neither. This design allows us to assess the model's effectiveness in dealing with class imbalance and partial class information, critical aspects in real-world applications where similar conditions are often encountered.

**Active Learning Experiment.** Tables 4 and 5 detail the parameter settings for tabular and image data, respectively. The tabular dataset experiments were based on the approach used in Chhabra et al. (2024) , with parameter settings derived from their open-source code. The experimental plots in their Appendix G guided our choices, and we adopted similar parameter scales. For image classification, we followed the design framework proposed in Kirsch et al. (2023), aligning with the parameter scales outlined in their Appendix E. Additionally, Table 6 describes the neural network architectures employed for *MNIST* and *EMNIST*.

**Table 4:** Active Learning Experiment Configuration for Tabular Datasets

| Attribute | Adult | CelebA | Jigsaw Toxicity |
|---|---|---|---|
| Number of Classes | 2 | 2 | 2 |
| Rounds | 12 | 11 | 8 |
| Initial Pool | 300 | 300 | 180 |
| Unlabeled Pool Size | 5000 | 3000 | 5000 |
| Acquisition Size | 500 | 150 | 500 |

## E ADDITIONAL EXPERIMENTS

### E.1 ABLATION EXPERIMENTS

Here, we supplement ablation experiments to evaluate the impact of perturbations on model performance, taking the *Bank* dataset as a case study. Specifically, we introduce Gaussian noise to $b$

**Table 5:** Active Learning Experiment Configuration for Image Datasets

| Attribute | MNIST | EMNIST | CIFAR10 |
|---|---|---|---|
| Number of Classes | 10 | 37 | 10 |
| Rounds | 14 | 10 | 9 |
| Initial Pool | 60 | 600 | 6000 |
| Unlabeled Pool Size | 420 | 2000 | 10000 |
| Acquisition Size | 30 | 100 | 2000 |

**Table 6:** Architecture of MNIST CNN

| Layer Type | Activation | Output Dimensions (incl. Padding) |
|---|---|---|
| Conv2d | ReLU | (32, 14, 14), P=1 |
| Conv2d | ReLU | (64, 7, 7), P=1 |
| Dropout | - | - |
| Linear | ReLU | 256 |
| Linear | - | num_classes |

samples in the training set with the highest and the lowest FI values, respectively, where $b$ takes values from [50, 100, 150, 200, 250, 300]. After introducing the perturbations, we evaluate the model's performance on the test set. The results are presented in Table 7 below. It can be observed that samples with higher FI values are more sensitive to perturbations, as their accuracy decreases more significantly compared to samples with lower FI values.

**Table 7:** Impact of Perturbations on Model Accuracy for Data Trimming.

| $b$ | Acc. after Perturbations (Top $b$-FI Samples) | Change in Acc. | Acc. after Perturbations (Bottom $b$-FI Samples) | Change in Acc. |
|---|---|---|---|---|
| 0 | 78.36% | / | 78.36% | / |
| 50 | 77.59% | -0.77% | 78.49% | 0.14% |
| 100 | 77.19% | -1.17% | 78.33% | -0.03% |
| 150 | 77.07% | -1.29% | 77.45% | -0.91% |
| 200 | 75.47% | -2.89% | 77.42% | -0.93% |
| 250 | 75.42% | -2.93% | 77.19% | -1.17% |
| 300 | 75.45% | -2.91% | 77.04% | -1.31% |

To address the trade-off between computational cost and performance, we add ablation experiments on two datasets. As shown in Table 8, the approximate method is over three times faster than the exact computation, with minimal impact on performance. It even achieves better results on the CelebA dataset. Additionally, as the dataset size increases, the proportion of time spent on training and prediction with the random forest decreases, making the speedup even more significant.

### E.2 ADDITIONAL EXPERIMENTS FOR DATA TRIMMING

Here, we present experimental results on four real-world datasets with noise in Figure 8, and nonlinear experimental results in Figure 9.

**Table 8:** Comparison of Approximate and Exact Methods on Different Datasets

| Dataset | Approx. Accuracy (%) | Exact Accuracy (%) | Approx. Time (s) | Exact Time (s) |
|---|---|---|---|---|
| MNIST | 96.0 | 96.2 | 396 | 1225 |
| CelebA | 83.2 | 81.5 | 260 | 1054 |

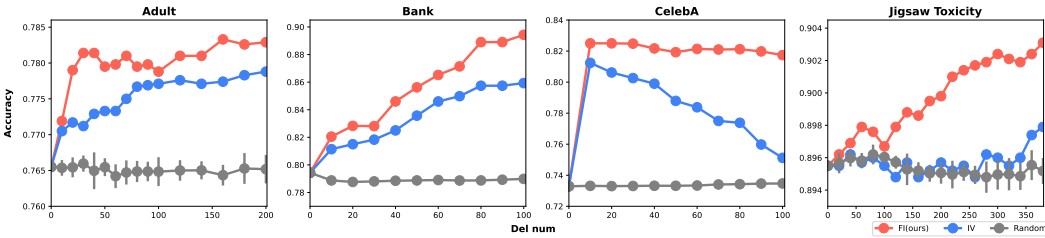

**Figure 8:** Accuracy curves of three data trimming methods on test sets of *Adult*, *Bank*, *CelebA* and *Jigsaw Toxicity*.

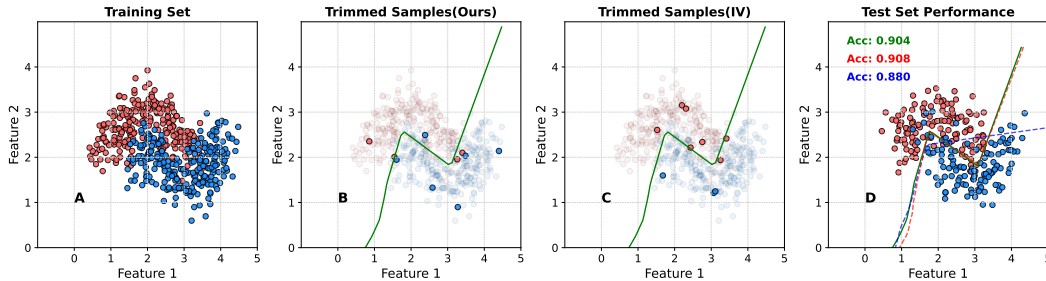

**Figure 9: Performance under Nonlinear Model.** Different colored points represent different classes. **A** shows the training set. **B** and **C** respectively denote the samples to be trimmed by FI method and IV method. **D** denotes test set. **Green** line: boundary without trimming. **Red** line: boundary after FI trimming. **Blue** line: boundary after IV trimming.

### E.3 Additional Experiments for Active Learning

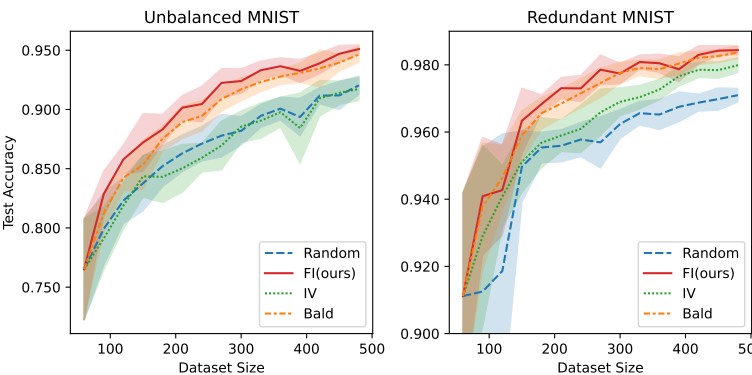

**Figure 10:** Classification performance due to the different active learning methods on *Unbalanced MNIST*, and *Redundant MNIST*.

The AG News dataset (Zhang et al., 2015) is a widely used benchmark in natural language processing (NLP), while the Office-31 dataset (Saenko et al., 2010) presents a more complex challenge due to its diverse domains. As shown in Figure 11, our method achieves consistently strong performance on both the AG News and Office-31 datasets, demonstrating its effectiveness across different types of classification tasks.

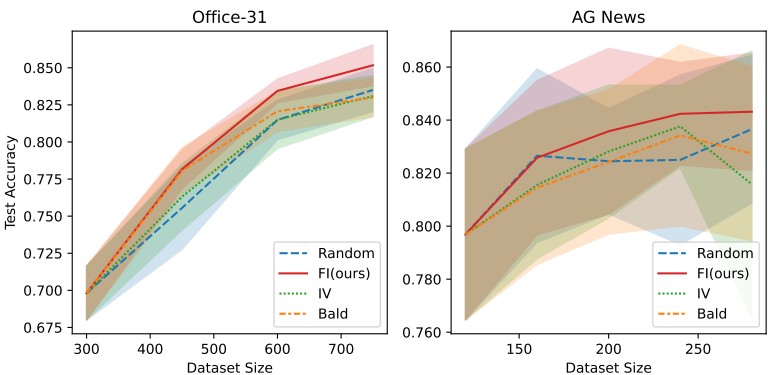

**Figure 11:** Classification performance due to the different active learning methods on *Office-31*, and *AGNews*.

**Table 9:** Time (seconds) for active learning algorithms over toy case on *MNIST*, *EMNIST*, and *CIFAR-10*.

| Methods | MNIST | EMNIST | CIFAR-10 |
|---------|-------|--------|----------|
| FI(ours) | 19 | 45 | 372 |
| IV | 13 | 15 | 204 |
| BALD | 9 | 15 | 214 |

### E.4 ADDITIONAL EXPERIMENTS FOR COMPARISON WITH MASKING EFFECT AND DATA SHAPLEY

**Masking Effect** is a key concept in model diagnostics, referring to the phenomenon where the impact of one outlier may be obscured by a nearby outlier, making detection difficult. Algorithms like FI, which are based on single perturbations, inevitably face interference from the masking effect when addressing model diagnostic issues. However, the primary focus of this research is on enhancing the predictive performance of the model, suggesting that the masking effect is unlikely to have a significant impact on the process.

To address this, an algorithm that accounts for the masking effect has been developed, referred to as FI-paired. This approach is inspired by Huang et al. (2007), where perturbations are applied simultaneously to two samples, and the corresponding FI values for the sample pairs are calculated. Subsequently, all sample pairs are sorted based on their FI values, and points are removed in descending order of FI. Given that the masking effect arises from outlier detection issues, which differ significantly from the context of active learning, comparisons are made solely regarding the data trimming problem, as illustrated in Figure 12.

The results indicate that, across both the *Adult* and *Bank* datasets, the performance of FI significantly surpasses that of FI-paired. Furthermore, the requirement to compute FI for each sample pair substantially increases computational costs. Therefore, in this context, utilizing FI directly is the more effective choice.

**Data Shapley** is indeed a well-known metric for data valuation (Ghorbani & Zou, 2019) and is applicable in this context. However, based on our experimental results, Shapley value-based methods do not demonstrate any advantage over our approach or IV. Wang et al. (2024) share a similar perspective on applying Data Shapley to data selection tasks, noting that its performance can be

comparable to random selection. Additionally, their high computational cost presents a significant drawback. Specifically:

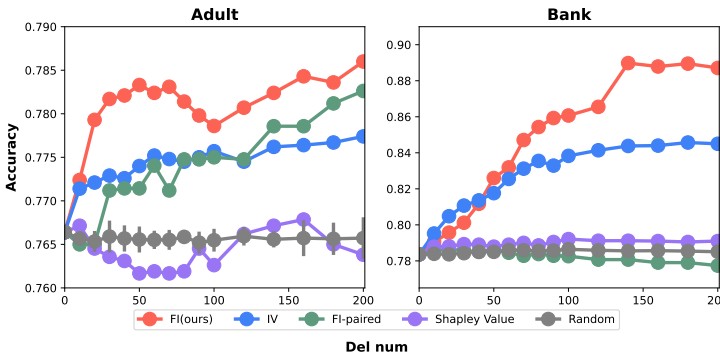

**Figure 12:** Performance comparison of different methods on *Adult* and *Bank*, including the newly added Masking Effect and Shapley method.

- In our experiments on data trimming with *Adult* and *Bank*, we observe that there is almost no significant difference between the performance of Data Shapley method and Random selection, with a slight improvement on *Bank* and even worse on *Adult* (see Figure 12).

- In our experiments on active learning with *MNIST*, we found that Data Shapley performed the worst, even slightly underperforming compared to Random selection (see Figure 13).

- In terms of time consumption, Data Shapley method takes approximately 860 times longer than FI.

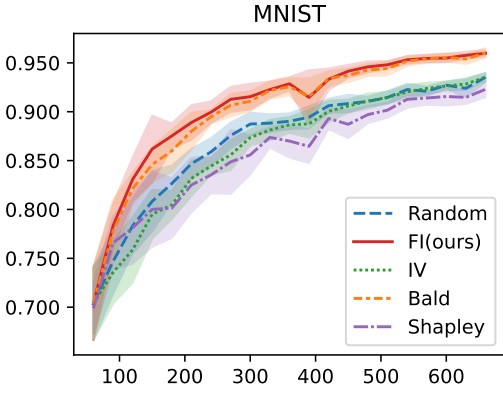

**Figure 13:** Performance comparison of different methods on *MNIST*, including the newly added Shapley method.

### E.5    K-FOLD CROSS-VALIDATION FOR DATA TRIMMING

Although all the datasets used in our experiments have clearly defined training and test sets, we further conduct 10-fold cross-validation on Data Trimming to verify the robustness of our method across different splits of the data.

In this experimental setup, the initial training set was randomly divided into 10 equally-sized subsets. In each iteration, one subset was held out as the validation set, while the remaining 9 subsets were used as the training set. The test set was fixed and remained the same as in the previous experiments, ensuring consistency across all evaluations. The results of 10-fold experiments are detailed in Figure 14, where we compare the overall performance and stability of our approach FI against the IV method.

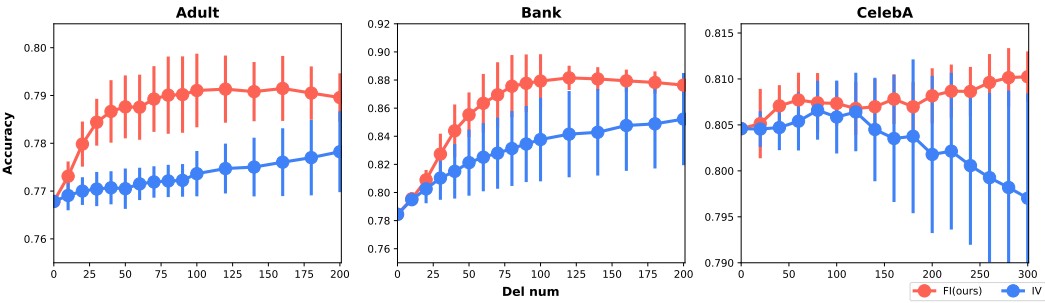

**Figure 14:** K-Fold Cross-Validation for Data Trimming: Performance Comparison of FI and IV on *Adult*, *Bank* and *CelebA*.

