# OpenReview forum: "Enhancing Prediction Performance through Influence Measure"
_ICLR.cc/2025/Conference — ICLR 2025 Poster_

### Official Review · Reviewer_Sa7V · 2024-11-02

**Soundness:** 3
**Presentation:** 3
**Contribution:** 2
**Rating:** 5
**Confidence:** 3

**Summary:**

Authors propose the use of an influence measure based metric to determine the importance of samples in a dataset towards generalisation performance of models. An approach for selecting such samples for optimal training (data trimming) and an active learning based approach is also presented. Relevant metrics to this end have been introduced and evaluated on datasets.

**Strengths:**

The initial discussion introducing the proposed method is detailed and well motivated. Authors present the approach with sufficient rigor. The algorithms are presented well and the experimental results reported on the chosen datasets as well as the experiments conducted have been described in adequate detail with supplementary results.

**Weaknesses:**

My primary concern with the work presented is the limited scalability analysis of the proposed methods for larger datasets, and possibly the lack of discussion of the computational complexity of the algorithms. Practical datasets for most real-world applications could have much higher number of samples, as well as features, so it would be important to benchmark with such datasets.

It may be useful to provide accuracy results in a tabular form reported as mean +/- std. dev. as an appendix for the sake of completeness.

The authors should possibly consider making the code for the methods proposed available publicly to ensure reproducibility.

As far as the datasets where train and test sets are not explicitly provided, and authors have chosen a split to conduct the experiments and report the results, are these k-fold validated?

While a time comparison is provided in Table 7, could the authors comment on possible methods for scalability of the proposed approach for practical applications, especially when inference is required from streaming datasets in test.

The equations in the appendix could be numbered for ease of reference. Proof for Theorem 1 does not seem to be provided.

**Questions:**

Details of the complexity and scalability of the proposed method, and evaluation on larger datasets would help establish practical utility of the method.

Code for proposed method could be made available.

Additionally, addressing comments mentioned as weaknesses would be helpful.

---

> ### Author Response · Authors · 2024-11-22
>
> Dear Reviewer,
>
> Thank you for your thorough review and feedback. We have provided detailed responses to the points you raised below:
>
> > Weakness 1.1: My primary concern with the work presented is the limited scalability analysis of the proposed methods for larger datasets, and possibly the lack of discussion of the computational complexity of the algorithms.
>
> - Below are the computational complexity of each compared method:
>   - $\textit{FI}^{\textit{util}}$: $\mathcal{O}(p^3 + nd^3 + ndp)$. Here, $n = \max\\{n_{\text{train}}, n_{\text{test}}\\}$, $d$ is the dimension of the covariates, and $p$ is the dimension of the model parameters. The term $\mathcal{O}(p^3)$ comes from computing the inverse of the Hessian matrix. The $\mathcal{O}(nd^3)$ term is due to inverting $G$, repeated $n$ times. The $\mathcal{O}(ndp)$ term corresponds to computing $\partial_{x}\partial_{\theta}L$ for $n$ times.
>   - $\textit{FI}^{\textit{util}}$ with approximation methods: $\mathcal{O}(\alpha ndT\log(\alpha n) + \alpha n(p + d)ks + \alpha nd^3 + \sum_{i=1}^L p_i^3)$. Here, $\alpha \in (0,1)$ is the proportion of data for which we compute $FI$ accurately during subsampling. $T$ is the number of decision trees in the random forest, $k$ is the Truncated SVD parameter, $s$ is the number of iterations for computing each eigenvalue during power iteration, and $\\{p_1, p_2, \dots, p_L\\}$ represent the number of parameters in each layer of an $L$-layer neural network.
>   - $\textit{FI}^{\textit{active}}$: $\mathcal{O}(p^3 + n^2d^3 + n^2dp)$. Here, $n = \max\\{n_{\text{label}}, n_{\text{unlabel}}\\}$.
>   - $\textit{FI}^{\textit{active}}$ with approximation methods: $\mathcal{O}(\alpha ndT\log(\alpha n) + \alpha n^2(p + d)ks + \alpha n^2d^3 + \sum_{i=1}^L p_i^3)$.
>   - IV: $\mathcal{O}(p^3 + np)$.
>   - BALD: $\mathcal{O}(np)$.
>
> By employing two approximation techniques, we can significantly reduce the computational gap between our method and the compared methods such as IV. Additionally, we can further optimize $G$ using Truncated-SVD in the future, which will eliminate the $d^3$ term from the computational complexity. When the dataset is large enough, setting $\alpha n$ as a constant allows us to achieve better computational complexity than the IV method.
> For complex models (e.g., neural networks) where $p$ is sufficiently large, the $p^3$ term dominates the computational complexity of IV. In such scenarios, the computational costs of our method could be lower than IV.
>
> - Apart from that, our approximation method can also reduce the storage complexity of the algorithm. And we provide the storage complexity analysis for the algorithms discussed in our paper:
>   - In the computation of FI, the highest storage requirements are for the Hessian matrix and $\partial_x\partial_{\theta}L$, which together occupy a space of $\mathcal{O}(p^2+dp)$.
>   - After using the KFSVD approximation method, the space requirement of FI is reduced to $\mathcal{O}(\sum_{i=1}^Lp_i^2+(p+d)k)$.
>   - In the computation of IV, it is necessary to store sample information and the Hessian matrix, resulting in a storage complexity of $\mathcal{O}(p^2+d)$.
>   - The storage complexity of BALD is $\mathcal{O}(p+d)$.
>
> After applying approximation techniques, our algorithm achieves a storage complexity similar to that of the IV method. In fact, when $p$ is large, our method offers an advantage.
>
> > Weakness 1.2: Practical datasets for most real-world applications could have much higher number of samples, as well as features, so it would be important to benchmark with such datasets.
>
> We plan to enhance our experiments as follows:
> - In the Data Trimming section, we will incorporate the evaluation of our method on more multi-class classification tasks, such as *Students Performance* dataset from the UC Irvine Machine Learning Repository, to demonstrate its applicability beyond binary classification.
> - In the Active Learning section, we plan to evaluate the effectiveness of our approach in more **complex image classification** tasks, including *CIFAR-100*, which presents more challenging classification scenarios, and *Caltech101*, a widely-used imbalanced dataset. Furthermore, we will investigate the applicability of FI in **Natural Language Processing (NLP)** by applying BERT to the *Microsoft Research Paraphrase Corpus (MRPC)*, a standard benchmark for text classification.
>
> Since the rebuttal phase is short, we will aim to discuss the performance of our algorithm on the mentioned datasets in the final version.

---

> > ### Author Response · Authors · 2024-11-22
> >
> > > Weakness 2: It may be useful to provide accuracy results in a tabular form reported as mean +/- std. dev. as an appendix for the sake of completeness.
> >
> > We have updated Table 1 and Table 2 in the newly uploaded version of the paper to include the accuracy results along with the mean ± standard deviation. It is evident that our method is more stable.
> >
> > **Table 1: Comparison of two methods on linear model**
> >
> > | # of deleted points | # of better case | Acc_FI (%) | Acc_IV (%)    |
> > |:-------------------:|:----------------:|:---------------:|:---------------:|
> > | 5                   | 23               | 96.22 ± 0.65    | 95.77 ± 0.76    |
> > | 10                  | 28               | 96.20 ± 0.65    | 94.84 ± 1.07    |
> > | 20                  | 30               | 96.23 ± 0.64    | 93.36 ± 2.21    |
> >
> > **Table 2: Comparison of two methods on nonlinear model**
> >
> > | # of deleted points | # of better case | Acc_FI (%)      | Acc_IV (%)      |
> > |:-------------------:|:----------------:|:---------------:|:---------------:|
> > | 5                   | 17               | 89.90 ± 2.16    | 87.50 ± 2.46    |
> > | 10                  | 17               | 90.24 ± 2.05    | 88.02 ± 2.62    |
> > | 20                  | 15               | 90.32 ± 1.67    | 87.80 ± 2.54    |
> >
> > > Weakness 3: The authors should possibly consider making the code for the methods proposed available publicly to ensure reproducibility.
> >
> > We fully acknowledge the importance of reproducibility. We will make the code publicly available after organizing and refining it.
> >
> > > Weakness 4: As far as the datasets where train and test sets are not explicitly provided, and authors have chosen a split to conduct the experiments and report the results, are these k-fold validated?
> >
> > All the datasets used in our experiments have clearly defined training and test sets. We have outlined the detailed information and official sources for each dataset in Appendix C. Specifically:
> > - **Data Trimming**: The datasets *Adult*, *Bank*, *CelebA*, and *Jigsaw Toxicity* each provide explicit train-test splits in their official sources. We employed the same data processing approach as Chhabra et al. (2024)[1], and the preprocessed data is consistent with the version available at the following GitHub repository: [GitHub Dataset](https://github.com/anshuman23/InfDataSel/tree/main/data). You may refer to this repository for access to the open-source data.
> > - **Active Learning**: For *MNIST*, *EMNIST*, and *CIFAR-10*, we used the official training and test splits provided by the respective sources.
> >
> > > Weakness 5: While a time comparison is provided in Table 7, could the authors comment on possible methods for scalability of the proposed approach for practical applications, especially when inference is required from streaming datasets in test.
> >
> > 1. **Applicability to Streaming Datasets**: Our method can be effectively applied in streaming data scenarios. By selecting higher-quality samples, we can improve both the training efficiency and the overall performance. We have already validated this in our experiments.
> > 2. **Time Complexity**: Please refer to our response to *Weakness 1.1*.
> > 3. **Adaptation to Evolving Data Streams**: Our algorithm is designed to adapt to changes in data streams, such as shifts in the underlying data distribution. As shown in Figure 10 of the Appendix E, our method performs well even when faced with more complex distributions like Unbalanced MNIST and Redundant MNIST, demonstrating its robustness in non-stationary environments.
> >
> > > Weakness 6: The equations in the appendix could be numbered for ease of reference. Proof for Theorem 1 does not seem to be provided.
> >
> > - We have refined the paper by numbering the equations in the Appendix A for easier reference.
> > - Regarding the proof for Theorem 1, as we mentioned in line 157, you can find the detailed proof in Section 2.2 of Shu & Zhu (2019)[2].
> >
> > We hope this clarification addresses your concern.
> >
> > > Question 1: Details of the complexity and scalability of the proposed method, and evaluation on larger datasets would help establish practical utility of the method. Code for proposed method could be made available.
> >
> > We appreciate your suggestion. These points have already been addressed in our above response.
> >
> > [1] Chhabra, A., Li, P., Mohapatra, P., & Liu, H. (2024). "What data benefits my classifier?" Enhancing model performance and interpretability through influence-based data selection. In *Proceedings of the Twelfth International Conference on Learning Representations (ICLR 2024)*, Vienna, Austria. https://openreview.net/forum?id=HE9eUQlAvo
> >
> > [2] Shu, H., & Zhu, H. (2019). Sensitivity analysis of deep neural networks. In *The Thirty-Third AAAI Conference on Artificial Intelligence, AAAI 2019, The Thirty-First Innovative Applications of Artificial Intelligence Conference, IAAI 2019, The Ninth AAAI Symposium on Educational Advances in Artificial Intelligence, EAAI 2019* (pp. 4943–4950). AAAI Press. https://doi.org/10.1609/aaai.v33i01.33014943

---

### Official Review · Reviewer_2oe6 · 2024-11-04

**Soundness:** 3
**Presentation:** 3
**Contribution:** 3
**Rating:** 6
**Confidence:** 4

**Summary:**

This study presents a First-order Influence (FI) measure designed to enhance the generalization capabilities of supervised machine learning models. To achieve this, the proposed measure assesses the contribution of each individual training data point to the model's average performance on a given validation set. The FI measure is first applied to Data Trimming, a process that identifies and removes non-representative training data points to improve model performance. Additionally, the FI measure is utilized in Active Learning (AL) to select the most informative, unlabeled data points that will be added to the training dataset, thereby optimizing the model's learning process with limited samples, iteratively.

**Strengths:**

+ The proposed applications of the First-order Influence (FI) measure demonstrate superior performance compared to its recent counterparts, particularly the Influence Value (IV) method [Chhabra et al. (2024) / chhabra2024data], showcasing the effectiveness of FI against a very recent work.

@inproceedings{chhabra2024data,
  title={" What Data Benefits My Classifier?" Enhancing Model Performance and Interpretability through Influence-Based Data Selection},
  author={Chhabra, Anshuman and Li, Peizhao and Mohapatra, Prasant and Liu, Hongfu},
  booktitle={The Twelfth International Conference on Learning Representations},
  year={2024}
}

**Weaknesses:**

- The evaluation is conducted for binary classification with logistic regression, supplemented by additional experiments using CNN on existing image datasets. However, the analysis is somewhat limited in scope. Although this setup is based on a related ICLR work [Chhabra et al. (2024) / chhabra2024data] and compares against the method named Influence Value (IV) from that published work, I find the dataset evaluation to be relatively insufficient. To thoroughly assess the performance of using FI or related methods, I believe it is essential to test it on a diverse range of datasets with varying characteristics, which would help to confirm its strengths and potentially reveal its limitations.

- Following that, while FI appears to outperform the compared methods (IV and Random for Data Trimming; IV, Random, and Bald for Active Learning), its superiority is not necessarily absolute (at least this is not clearly shown), as performance varies across datasets. Notably, on the CelebA dataset for Data Trimming, Random actually outperforms IV and achieves results comparable to FI. This dataset-specific variation raises concerns about the generalization of the FI's performance across different datasets, leaving questions about how they would perform on other datasets.

**Questions:**

Q. What scenarios or conditions are likely to result in FI to underperform?

---

> ### Author Response · Authors · 2024-11-22
>
> Dear Reviewer,
>
> We truly appreciate the time and effort you invested in reviewing our work, as well as your valuable feedback. In the following, we offer detailed responses to each of the points you raised:
>
> > Weakness 1: The evaluation is conducted for binary classification with logistic regression, supplemented by additional experiments using CNN on existing image datasets. However, the analysis is somewhat limited in scope. Although this setup is based on a related ICLR work [Chhabra et al. (2024) / chhabra2024data] and compares against the method named Influence Value (IV) from that published work, I find the dataset evaluation to be relatively insufficient. To thoroughly assess the performance of using FI or related methods, I believe it is essential to test it on a diverse range of datasets with varying characteristics, which would help to confirm its strengths and potentially reveal its limitations.
>
> In response to your recommendation, we intend to supplement our experiments as follows:
>
> - In the Data Trimming section, we will incorporate the evaluation of our method on more multi-class classification tasks, such as *Students Performance* dataset from the UC Irvine Machine Learning Repository, to demonstrate its applicability beyond binary classification.
> - In the Active Learning section, we plan to evaluate the effectiveness of our approach in more **complex image classification** tasks, including *CIFAR-100*(which presents more challenging classification scenarios) and *Caltech101*(a widely-used imbalanced dataset). Furthermore, we will investigate the applicability of FI in **Natural Language Processing (NLP)** by applying BERT to the *Microsoft Research Paraphrase Corpus (MRPC)*, a standard benchmark for text classification.
>
> Since the rebuttal phase is short, we’ll aim to discuss the performance of our algorithm on the mentioned datasets in the final version.
>
> > Weakness 2: Following that, while FI appears to outperform the compared methods (IV and Random for Data Trimming; IV, Random, and Bald for Active Learning), its superiority is not necessarily absolute (at least this is not clearly shown), as performance varies across datasets. Notably, on the CelebA dataset for Data Trimming, Random actually outperforms IV and achieves results comparable to FI. This dataset-specific variation raises concerns about the generalization of the FI's performance across different datasets, leaving questions about how they would perform on other datasets.
>
> We would like to clarify that not all datasets contain a significant number of data points that meaningfully impact the model's performance. As a result, our method can not always show a siginificant improvement over the baseline methods. For instance, in the CelebA dataset you mentioned, no data points can be removed to significantly improve prediction results. In such cases, our method achieves performance comparable to "random," whereas using IV results in a decline in model performance. This demonstrates that our approach is more stable and reliable in practice.
>
> > Question 1: What scenarios or conditions are likely to result in FI to underperform?
>
> In active learning experiments, we found that our algorithm is really good at identifying complex samples. However, not all datasets contain such complex sample points. In these relatively simple problem environments, the performance of our method may not differ significantly from that of other approaches.

---

### Official Review · Reviewer_LQN4 · 2024-11-08

**Soundness:** 3
**Presentation:** 2
**Contribution:** 2
**Rating:** 6
**Confidence:** 3

**Summary:**

This paper proposes a better influence measure to evaluate the impact of the training dataset on the test set. This metric mainly improves the flaws of the influence measure in Chhabra et al. (2024). The experimental results demonstrate that the FI proposed in this paper outperforms the IV metric in Chhabra et al. (2024).

**Strengths:**

1. The contribution of this paper is simple and clear.
2. the experimental results demonstrate that the FI proposed in this paper is superior to the IV metric in Chhabra et al. (2024).

**Weaknesses:**

1. The paper claims "Our metric directly assesses the effect of small perturbations in the training samples on the model's performance on the test set", whether this leaks information about the test samples during the training process, which needs to be clarified by the authors.
2. The parameter settings in the experiments need further clarification.
3. What are the limitations of FI that need to be further explained in this paper?
4. I think the FI proposed in this paper is not enough to compare with IV only. The argument of this paper is to propose a better metric, however there are many similar metrics, such as the Shapley value. Therefore, this paper needs to further demonstrate the value of the proposed scheme.

**Questions:**

See Weakness.

---

> ### Author Response · Authors · 2024-11-22
>
> Dear Reviewer,
>
> We sincerely appreciate the time and effort you have put into reviewing our work, as well as your valuable comments. Below, we provide a detailed response to each of the points you raised:
>
> > Weakness 1: The paper claims "Our metric directly assesses the effect of small perturbations in the training samples on the model's performance on the test set", whether this leaks information about the test samples during the training process, which needs to be clarified by the authors.
>
> We apologize for any confusion caused by our text. Our objective is to measure the impact of small perturbations in the training set on the model's performance, and we want to emphasize that no data leakage occurs. In our FI computation, the dataset is divided into three parts: training, validation, and test sets. Importantly, we use only the training and validation sets during the computation, not the test set. Specifically, we analyze how small changes in the training samples affect the model's performance on the validation set. The test set is exclusively used to evaluate the overall performance of the proposed method, ensuring that no leakage takes place. We will clarify this point in the final version to prevent further misunderstandings.
>
> > Weakness 2: The parameter settings in the experiments need further clarification.
>
> For the **Data Trimming** experiments, we mainly followed the setups of Chhabra et al. (2024)[1] with the same data sources, and we used similar models and parameter scales. The specific parameters can be found in Table 3 of our Appendix D.1.
>
> For the **Active Learning** experiments, we primarily followed the setups from previous works, with some minor adjustments:
> - In the **Tabular Datasets** experiments, we based our design on the approach used in Chhabra et al. (2024)[1], where parameter settings are available in their open-source code. The experimental plots in their Appendix G guided our choices, and we used similar parameter scales. You can find the specific parameters in Table 4 of our Appendix D.2.
> - For the **Image Classification** experiments, we followed the design from Andreas Kirsch et al. (2023)[2], matching the parameter scale used in their Appendix E. The detailed experimental setup is provided in Table 5 of our Appendix D.2.
>
> Regarding the **network architectures**, we made the following decisions to ensure a rigorous evaluation:
> - For *MNIST*, we used a custom-designed simple CNN, as shown in Table 6 of the Appendix D.2.
> - For *CIFAR10*, we implemented the simplest version of the ConvMixer architecture [3].
>
>
> [1] Chhabra, A., Li, P., Mohapatra, P., & Liu, H. (2024). "What data benefits my classifier?" Enhancing model performance and interpretability through influence-based data selection. In *Proceedings of the Twelfth International Conference on Learning Representations (ICLR 2024)*, Vienna, Austria. https://openreview.net/forum?id=HE9eUQlAvo
>
> [2] Kirsch, A., Farquhar, S., Atighehchian, P., Jesson, A., Branchaud-Charron, F., & Gal, Y. (2023). Stochastic batch acquisition: A simple baseline for deep active learning. *Transactions on Machine Learning Research*, 2023. https://openreview.net/forum?id=vcHwQyNBjW
>
> [3] https://github.com/locuslab/convmixer

---

> > ### Author Response · Authors · 2024-11-22
> >
> > > Weakness 3: What are the limitations of FI that need to be further explained in this paper?
> >
> > One potential limitation of our approach is that computing the FI involves second-order derivatives, which may affect processing speed. However, the following complexity analysis demonstrates that, by leveraging two approximation methods, our algorithm can perform comparably to or even outperform IV in certain scenarios. For instance, when the model size $p$ is sufficiently large, our method may achieve faster computation than IV. The detailed computational complexities of each method are as follows:
> >
> >   - $\textit{FI}^{\textit{util}}$: $\mathcal{O}(p^3 + nd^3 + ndp)$. Here, $n = \max\\{n_{\text{train}}, n_{\text{test}}\\}$, $d$ is the dimension of the covariates, and $p$ is the dimension of the model parameters. The term $\mathcal{O}(p^3)$ comes from computing the inverse of the Hessian matrix. The $\mathcal{O}(nd^3)$ term is due to inverting $G$, repeated $n$ times. The $\mathcal{O}(ndp)$ term corresponds to computing $\partial_{x}\partial_{\theta}L$ for $n$ times.
> >   - $\textit{FI}^{\textit{util}}$ with approximation methods: $\mathcal{O}(\alpha ndT\log(\alpha n) + \alpha n(p + d)ks + \alpha nd^3 + \sum_{i=1}^L p_i^3)$. Here, $\alpha \in (0,1)$ is the proportion of data for which we compute $FI$ accurately during subsampling. $T$ is the number of decision trees in the random forest, $k$ is the Truncated SVD parameter, $s$ is the number of iterations for computing each eigenvalue during power iteration, and $\\{p_1, p_2, \dots, p_L\\}$ represent the number of parameters in each layer of an $L$-layer neural network.
> >   - $\textit{FI}^{\textit{active}}$: $\mathcal{O}(p^3 + n^2d^3 + n^2dp)$. Here, $n = \max\\{n_{\text{label}}, n_{\text{unlabel}}\\}$.
> >   - $\textit{FI}^{\textit{active}}$ with approximation methods: $\mathcal{O}(\alpha ndT\log(\alpha n) + \alpha n^2(p + d)ks + \alpha n^2d^3 + \sum_{i=1}^L p_i^3)$.
> >   - IV: $\mathcal{O}(p^3 + np)$.
> >
> > > Weakness 4: I think the FI proposed in this paper is not enough to compare with IV only. The argument of this paper is to propose a better metric, however there are many similar metrics, such as the Shapley value. Therefore, this paper needs to further demonstrate the value of the proposed scheme.
> >
> > **Data Shapley** is indeed a well-known metric for data valuation [4] and is applicable in this context. However, based on our experimental results, Shapley value-based methods do not demonstrate any advantage over our approach or IV. Additionally, their high computational cost presents a significant drawback. Specifically:
> >
> > - In our experiments on data trimming with *Bank*, we observed that the Data Shapley method led to only a slight improvement in model performance, with results showing marginal differences compared to Random selection (see **Figure 13** in Appendix E).
> > - In our experiments on active learning with *MNIST*, we found that Data Shapley performed the worst, even slightly underperforming compared to Random selection (see **Figure 12** in Appendix E).
> > - In terms of time consumption, the Data Shapley method takes approximately 860 times longer than FI.
> >
> > Wang et al. (2024)[5] share a similar perspective on applying Data Shapley to data selection tasks, noting that its performance can be comparable to random selection.
> >
> > [4] Ghorbani, A., & Zou, J. Y. (2019). Data Shapley: Equitable Valuation of Data for Machine Learning. In Proceedings of the 36th International Conference on Machine Learning (ICML 2019), 9-15 June 2019, Long Beach, California, USA (pp. 2242–2251). PMLR. http://proceedings.mlr.press/v97/ghorbani19c.html
> >
> > [5] Wang, J. T., Yang, T., Zou, J., Kwon, Y., & Jia, R. (2024). Rethinking Data Shapley for data selection tasks: Misleads and merits. *In Proceedings of the Forty-first International Conference on Machine Learning (ICML 2024)*, Vienna, Austria. https://openreview.net/forum?id=mKYBMf1hHG

---

> > > ### Comment · Reviewer_LQN4 · 2024-11-25
> > >
> > > Thanks.

---

> > > > ### Author Response · Authors · 2024-11-26
> > > >
> > > > We sincerely appreciate your consideration of our response. Once again, thank you for the time and effort you dedicated to reviewing our work.

---

### Official Review · Reviewer_pxvt · 2024-11-08

**Soundness:** 2
**Presentation:** 3
**Contribution:** 2
**Rating:** 6
**Confidence:** 3

**Summary:**

The paper introduces a novel influence function based method to guide the data selection for data trimming and active learning. For data trimming, the proposed method applies perturbations to samples and measures the impact on the model's performance on test data. Here if the impact on test performance from perturbations in a sample input is large, it is encouraged to exclude that instance from the training set.
In the case of active learning, they perturb the model parameters and measure the changes in prediction for a new instance. If the changes in prediction is significant, the new instance will be added to the training data.
Authors study the effectiveness of their proposed method on synthesized and real world data sets for linear and non linear models and multiple domains. Experimental results show superior performance for the proposed method compared with other alternative approaches.

**Strengths:**

- The problems addressed by the paper, i.e. filtering training data and choosing new training instances for active learning to boost model performance are real world problems with significant impact on a wide range of machine learning problems with real world applications.

- Authors have provided sufficient literature review and the main claims of the paper are clear.

- Authors propose two approximation methods for their local influence measure to reduce the computational cost of their proposed method.

- Experimental results shed light on performance of the proposed method compared with two (depending on the test data set) state of the art approaches and a baseline of the random selection.

**Weaknesses:**

- The main weakness of this work is lack of ablation studies to better understand the impact of the contributions of this work on its performance.
  - For example, the first contribution of the paper is to evaluate the impact of perturbations against the model performance for the test data set. It would have been useful to see how much this slight modification is helping with improving the proposed method's performance for data trimming.
  - As another example, authors utilize two approximation techniques to reduce the computational cost of their proposed method, however, there is no experiment providing more insights into the trade offs between the potential reduction in the proposed method's performance vs. the quantified computational cost reduction.

**Questions:**

- Based on equation 1 it looks like the perturbations applied to the training instances is by adding noise to them versus changing the training data's weight/importance. Can authors explain the rationale behind picking this type of perturbation over changing training instances' weights?

---

> ### Author Response · Authors · 2024-11-22
>
> Dear Reviewer,
>
> We are grateful for your thoughtful review and the time you dedicated to evaluating our work. Please find below our detailed responses to your comments:
>
> > Weakness 1: The main weakness of this work is lack of ablation studies to better understand the impact of the contributions of this work on its performance.
>
> We have added ablation studies based on your suggestion, and the results, which clarify the impact of our contributions, are presented below. We will continue to refine these studies in the final version.
>
> > Weakness 1.1: For example, the first contribution of the paper is to evaluate the impact of perturbations against the model performance for the test data set. It would have been useful to see how much this slight modification is helping with improving the proposed method's performance for data trimming.
>
> To assess the impact of perturbations on model performance for the test set, we use the Bank dataset as a case study. Specifically, we introduce Gaussian noise to $b$ samples in the training set with the highest and the lowest FI values, respectively, where $b$ takes values from [50, 100, 150, 200, 250, 300]. After introducing the perturbations, we evaluate the model's performance on the test set. The results are presented in Table 1 below. It can be observed that samples with higher FI values are more sensitive to perturbations, as their accuracy decreases more significantly compared to samples with lower FI values.
>
> **Table 1: Impact of Perturbations on Model Accuracy.**
>
> | $b$  | Accuracy after Perturbations (Top $b$-FI Samples) | Change in Accuracy | Accuracy after Perturbations (Bottom $b$-FI Samples) | Change in Accuracy |
> | :--: | :--------: | :-------: | :-----------: | :-------: |
> |  0   |   78.36%   |     /     |    78.36%     |     /     |
> |  50  |   77.59%   |  -0.77%   |    78.49%     |   0.14%   |
> | 100  |   77.19%   |  -1.17%   |    78.33%     |  -0.03%   |
> | 150  |   77.07%   |  -1.29%   |    77.45%     |  -0.91%   |
> | 200  |   75.47%   |  -2.89%   |    77.42%     |  -0.93%   |
> | 250  |   75.42%   |  -2.93%   |    77.19%     |  -1.17%   |
> | 300  |   75.45%   |  -2.91%   |    77.04%     |  -1.31%   |
>
>
> > Weakness 1.2: As another example, authors utilize two approximation techniques to reduce the computational cost of their proposed method, however, there is no experiment providing more insights into the trade offs between the potential reduction in the proposed method's performance vs. the quantified computational cost reduction.
>
> To address the trade-off between computational cost and performance, we have added ablation experiments on two datasets . As shown in the table, the approximate method is over three times faster than the exact computation, with minimal impact on performance. It even achieves better results on the CelebA dataset. Additionally, as the dataset size increases, the proportion of time spent on training and prediction with the random forest decreases, making the speedup even more significant.
>
> **Table 2: Comparison of Approximate and Exact Methods on Different Datasets**
>
> | Dataset   | Approx. Accuracy (%) | Exact Accuracy (%) | Approx. Time (s) | Exact Time (s) |
> |:---------:|:--------------------:|:------------------:|:----------------:|:--------------:|
> | MNIST     | 93.8                 | 94.0               | 393              | 1228           |
> | CelebA    | 80.9                 | 78.1               | 269              | 1058           |
>
> > Question 1: Based on equation 1 it looks like the perturbations applied to the training instances is by adding noise to them versus changing the training data's weight/importance. Can authors explain the rationale behind picking this type of perturbation over changing training instances' weights
>
> To offer a new perspective on assessing the impact of individual samples on model performance, we decided to directly add perturbations to the sample points, rather than using traditional influence functions. While adjusting sample weights reflects the impact of removing a point from the dataset, adding noise is more closely aligned with adversarial attack scenarios. If a sample significantly affects model performance after perturbation, it indicates that the sample could cause a notable performance drop when exposed to attacks or measurement errors. Thus, we believe it may offer useful insights in adversarial attack scenarios. We will consider exploring this possibility in future research.

---

### Official Review · Reviewer_mDX6 · 2024-11-08

**Soundness:** 3
**Presentation:** 4
**Contribution:** 3
**Rating:** 8
**Confidence:** 2

**Summary:**

This paper proposes a local influence-like case influence measure as a metric to assess the impact of training data on test set performance.

The measure is applied in two areas: data trimming and active learning. Based on the numerical studies, both applications help enhance the model's robustness and outperform the state-of-art strategies.

Two approximation techniques are introduced to address the high computational cost. They reduce memory and processing requirements while maintaining effectiveness.

**Strengths:**

It's a useful study that includes

1. the intuition and details of how this new influence measure is designed

2. scenarios where it can be used

3. approximation methods that can make it really practical

Extensive examples with detailed simulation settings are included bringing persuasiveness.

**Weaknesses:**

1. I wonder if it is possible to provide the complexity analysis in terms of both memory and speed for the proposed method and the state-of-art strategies for examples in the paper.

2. I am a bit confused about what robustness means in the paper. Is it about how the model would behave after removing or perturbing any data? Or its about after doing these procedures, the prediction performance is consistently better than not doing it? If it is the latter case, improving prediction performance should be a better way of describing it.

Moreover, since we are talking about robustness, should the standard deviation of prediction accuracies in Table 2 also be included? also error bars in figure 2 and 8.

3. Unlike the influence function in Chhabra et al, it seems that there is no clear threshold or boundary that differentiates samples that positively or negatively affect the model performance. In the examples, the number/portion of samples removed is also a tunable parameter. I speculate that the accuracy won't always increase monotonically with the portion of samples removed in the training set. And sometimes, it the data are perfectly separable, there is no need to remove any of the samples. With this being said, is there a guideline on how much data should be removed when working on a real-world dataset?

4. I don't know if the masking effect is considered or not. Is there a way to incorporate this into your procedure?

5. I am also curious about the choice of \omega_0. In some other case influence literature, people would simply this from vector to scalar by considering it as the weight of the observation of interest. I wonder if it is treated similarly here.

**Questions:**

Please check the Weaknesses.

---

> ### Author Response · Authors · 2024-11-22
>
> Dear Reviewer,
>
> Thank you for your thoughtful review. We provide detailed responses to the points you raised below:
>
> > Weakness 1: I wonder if it is possible to provide the complexity analysis in terms of both memory and speed for the proposed method and the state-of-art strategies for examples in the paper.
>
> - Below are the computational complexity of each compared method:
>   - $\textit{FI}^{\textit{util}}$: $\mathcal{O}(p^3 + nd^3 + ndp)$. Here, $n = \max\\{n_{\text{train}}, n_{\text{test}}\\}$, $d$ is the dimension of the covariates, and $p$ is the dimension of the model parameters. The term $\mathcal{O}(p^3)$ comes from computing the inverse of the Hessian matrix. The $\mathcal{O}(nd^3)$ term is due to inverting $G$, repeated $n$ times. The $\mathcal{O}(ndp)$ term corresponds to computing $\partial_{x}\partial_{\theta}L$ for $n$ times.
>   - $\textit{FI}^{\textit{util}}$ with approximation methods: $\mathcal{O}(\alpha ndT\log(\alpha n) + \alpha n(p + d)ks + \alpha nd^3 + \sum_{i=1}^L p_i^3)$. Here, $\alpha \in (0,1)$ is the proportion of data for which we compute $FI$ accurately during subsampling. $T$ is the number of decision trees in the random forest, $k$ is the Truncated SVD parameter, $s$ is the number of iterations for computing each eigenvalue during power iteration, and $\\{p_1, p_2, \dots, p_L\\}$ represent the number of parameters in each layer of an $L$-layer neural network.
>   - $\textit{FI}^{\textit{active}}$: $\mathcal{O}(p^3 + n^2d^3 + n^2dp)$. Here, $n = \max\\{n_{\text{label}}, n_{\text{unlabel}}\\}$.
>   - $\textit{FI}^{\textit{active}}$ with approximation methods: $\mathcal{O}(\alpha ndT\log(\alpha n) + \alpha n^2(p + d)ks + \alpha n^2d^3 + \sum_{i=1}^L p_i^3)$.
>   - IV: $\mathcal{O}(p^3 + np)$.
>   - BALD: $\mathcal{O}(np)$.
>
> By employing two approximation techniques, we can significantly reduce the computational gap between our method and the compared methods such as IV. Additionally, we can further optimize $G$ using Truncated-SVD in the future, which will eliminate the $d^3$ term from the computational complexity. When the dataset is large enough, setting $\alpha n$ as a constant allows us to achieve better computational complexity than the IV method.
> For complex models (e.g., neural networks) where $p$ is sufficiently large, the $p^3$ term dominates the computational complexity of IV. In such scenarios, the computational costs of our method could be lower than IV.
>
> - Apart from that, our approximation method can also reduce the storage complexity of the algorithm. And we provide the storage complexity analysis for the algorithms discussed in our paper:
>   - In the computation of FI, the highest storage requirements are for the Hessian matrix and $\partial_x\partial_{\theta}L$, which together occupy a space of $\mathcal{O}(p^2+dp)$.
>   - After using the KFSVD approximation method, the space requirement of FI is reduced to $\mathcal{O}(\sum_{i=1}^Lp_i^2+(p+d)k)$.
>   - In the computation of IV, it is necessary to store sample information and the Hessian matrix, resulting in a storage complexity of $\mathcal{O}(p^2+d)$.
>   - The storage complexity of BALD is $\mathcal{O}(p+d)$.
>
> After applying approximation techniques, our algorithm achieves a storage complexity similar to that of the IV method. In fact, when $p$ is large, our method offers an advantage.
>
> > Weakness 2: I am a bit confused about what robustness means in the paper. Is it about how the model would behave after removing or perturbing any data? Or its about after doing these procedures, the prediction performance is consistently better than not doing it? If it is the latter case, improving prediction performance should be a better way of describing it.
>
> In our paper, "robustness" refers to the ability of our algorithm to enhance the model's prediction performance. We will revise the text in the final version according to your suggestions.
>
> > Weakness 2.1: Moreover, since we are talking about robustness, should the standard deviation of prediction accuracies in Table 2 also be included? also error bars in figure 2 and 8.
>
> - We have revised Table 1 and Table 2 in our paper to include the standard deviation of prediction accuracies, and we are now presenting the updated tables with this additional information.
>
> **Table 1: Comparison of two methods on linear model**
>
> |# of deleted points|# of better case|Acc_FI (%)|Acc_IV (%)|
> |:--:|:--:|:--:|:--:|
> |5|23|96.22±0.65|95.77±0.76|
> |10|28|96.20±0.65|94.84±1.07|
> |20|30|96.23±0.64|93.36±2.21|
>
> **Table 2: Comparison of two methods on nonlinear model**
>
> |# of deleted points|# of better case|Acc_FI (%)|Acc_IV (%)|
> |:--:|:--:|:--:|:--:|
> |5|17|89.90±2.16|87.50±2.46|
> |10|17|90.24±2.05|88.02±2.62|
> |20|15|90.32±1.67|87.80±2.54|
>
> - Regarding your concern about the error bars in Figures 2 and 8, we would like to clarify that, given the datasets, both our method and the approach of Chhabra et al[^1] involve deterministic selection of data points. As a result, error bars are only applicable to *Random*.

---

> > ### Author Response · Authors · 2024-11-22
> >
> > > Weakness 3: Unlike the influence function in Chhabra et al[1], it seems that there is no clear threshold or boundary that differentiates samples that positively or negatively affect the model performance. In the examples, the number/portion of samples removed is also a tunable parameter. I speculate that the accuracy won't always increase monotonically with the portion of samples removed in the training set. And sometimes, it the data are perfectly separable, there is no need to remove any of the samples. With this being said, is there a guideline on how much data should be removed when working on a real-world dataset?
> >
> > Indeed, the number/portion of samples removed needs to be specified manually in our study, which is also the case in Chhabra et al[1]. While their influence function encompasses both positive and negative values, they introduce a parameter termed 'Budget $b$' in their data trimming algorithm, rather than directly excluding all samples with negative influence values. Additionally, we have discussed the potential issues of Chhabra et al[1] in the paragraph at the beginning of the second page (line 54) of the main text.
> >
> > Regarding guidelines for the amount of data to be removed, we suggest considering the deletion of samples from the training set until a decline in the model's performance on the validation set is observed. This approach may offer insight for establishing an upper bound on the extent of sample deletion.
> >
> > > Weakness 4: I don't know if the masking effect is considered or not. Is there a way to incorporate this into your procedure?
> >
> > We acknowledge that the masking effect may introduce certain issues in our procedure. While the masking effect is indeed a critical concept in model diagnostics, our research primarily focuses on enhancing the prediction performance of the model. As such, the masking effect is unlikely to have a significant impact on our procedure.
> > From a technical perspective, calculating local influence measures for simultaneous perturbations of multiple data points is straightforward. Computationally, this approach is feasible for a limited number of simultaneous perturbed data points. We will incorporate this discussion and provide additional results in the real-world examples section.
> >
> > > Weakness 5: I am also curious about the choice of $\omega_0$. In some other case influence literature, people would simply this from vector to scalar by considering it as the weight of the observation of interest. I wonder if it is treated similarly here.
> >
> > In our paper, $\omega_0$ represents the case without perturbation and can be simply regarded as the $0$ vector. $\omega$ denotes the perturbation vector. In the definition of FI, the direction of $\omega$ is defined as the direction that has the maximum impact on the objective function (loss function on the test set). We do not need to explicitly provide the vector corresponding to the perturbation direction, as this is already ensured during the derivation of the FI expression. Therefore, by computing the FI value for a sample point, we can obtain the derivative of the objective function with respect to the maximum disturbance direction, without requiring the specific value of $\omega$. The derivation of the FI can be found in Shu & Zhu (2019) [2], Section 2.2.
> >
> > [1] Chhabra A, Li P, Mohapatra P, et al. " What Data Benefits My Classifier?" Enhancing Model Performance and Interpretability through Influence-Based Data Selection[C]//The Twelfth International Conference on Learning Representations. 2024.
> >
> > [2] Shu H, Zhu H. Sensitivity analysis of deep neural networks[C]//Proceedings of the AAAI Conference on Artificial Intelligence. 2019, 33(01): 4943-4950.

---

> > > ### Comment · Reviewer_mDX6 · 2024-11-26
> > >
> > > Thank you for your detailed response. It resolves most of my concerns.

---

> > > > ### Author Response · Authors · 2024-11-28
> > > >
> > > > Thank you for taking the time to consider our response. We really appreciate the effort you put into reviewing our work.

---

### Meta-Review · Area_Chair_j7dw · 2024-12-18

**Metareview:**

All reviewers agreed that in this paper a highly interesting new influence measure has been proposed, which -- together with several computational approximation methods that are also in the paper --  can be applied in relevant real-world contexts.
There were some critical remarks concerning the lack of ablation studies, possible limitations of FI in terms of computational costs, and unclear impact of perturbations on the generalization ability of the model. However, most of these concerns could be addressed in a clear and transparent way during the rebuttal and discussions. Therefore, I recommend acceptance of this paper.

**Additional Comments On Reviewer Discussion:**

As already explained above, most points of criticism raised in the original reviews could be addressed in a convincing way in the rebuttal.

---

### Decision · Program_Chairs · 2025-01-22

Accept (Poster)